# Patterned cortical tension mediated by N-cadherin controls cell geometric order in the *Drosophila* eye

Eunice HoYee Chan*[†], Pruthvi Chavadimane Shivakumar[†], Raphaël Clément, Edith Laugier, Pierre-François Lenne*

Aix Marseille Univ, CNRS, IBDM, Marseille, France

**Abstract** Adhesion molecules hold cells together but also couple cell membranes to a contractile actomyosin network, which limits the expansion of cell contacts. Despite their fundamental role in tissue morphogenesis and tissue homeostasis, how adhesion molecules control cell shapes and cell patterns in tissues remains unclear. Here we address this question in vivo using the *Drosophila* eye. We show that cone cell shapes depend little on adhesion bonds and mostly on contractile forces. However, N-cadherin has an indirect control on cell shape. At homotypic contacts, junctional N-cadherin bonds downregulate Myosin-II contractility. At heterotypic contacts with E-cadherin, unbound N-cadherin induces an asymmetric accumulation of Myosin-II, which leads to a highly contractile cell interface. Such differential regulation of contractility is essential for morphogenesis as loss of N-cadherin disrupts cell rearrangements. Our results establish a quantitative link between adhesion and contractility and reveal an unprecedented role of N-cadherin on cell shapes and cell arrangements.

*For correspondence: ho-yee.chan@univ-amu.fr (EHoYC); pierre-francois.lenne@univ-amu.fr (P-FL)

[†]These authors contributed equally to this work

Competing interests: The authors declare that no competing interests exist.

## Introduction

Cells acquire different shapes and arrangements to form tissues, depending on their functions and microenvironment. During tissue morphogenesis, cells actively form and remodel their cell contacts, generating forces to drive various morphogenetic events (*Lecuit and Lenne, 2007*). In epithelia, cell division (*Herszterg et al., 2013*; *Guillot and Lecuit, 2013*; *Founounou et al., 2013*), cell intercalation (*Bertet et al., 2004*; *Blankenship et al., 2006*) and cell delamination (*Marinari et al., 2012*) are basic mechanisms of morphogenesis, which all involve gain or loss of cell contacts (*Heisenberg and Bellaïche, 2013*). Two systems contribute to changes in cell contacts: Cadherin complexes and actomyosin networks (*Harris, 2012*; *Baum and Georgiou, 2011*).

At the level of a single cell contact, formation of cadherin-cadherin bonds favors contact expansion. Actomyosin contractility acts antagonistically by reducing cell contact size (*Lecuit and Lenne, 2007*; *Winklbauer, 2015*). There is numerous evidence in vivo that shows actomyosin-generated tension regulates cell shape (*Rauzi et al., 2008*; *Martin et al., 2009*). In vitro, contact size is mainly determined by actomyosin contractility but not cadherin engagement (*Maître et al., 2012*). However, in *Drosophila* retina, N-cadherin mutants show drastic alteration of contact size and cell shape (*Hayashi and Carthew, 2004*), which suggests that cadherin-associated adhesion cannot be discounted. Even though the forces produced by cadherins and actomyosin networks act antagonistically, both systems are interconnected as cadherins are associated with intracellular actomyosin networks via catenins and other actin-binding proteins (*Priya et al., 2013*; *Röper, 2015*).

Due to the intrinsic links between cadherin-dependent adhesion and actomyosin contractility, it is challenging to address whether and how cadherin adhesion regulates cell shape. What is the direct contribution of cadherin-cadherin bonds to cell shape? Do cadherins influence cell shape through

actomyosin contractility? To address these questions, we investigated the origin of cell shapes in vivo in the highly organized *Drosophila* retina, which features differential expression of cadherin molecules and is amenable to quantification of cell shapes and mechanical measurements. In particular, the *Drosophila* retina is an ideal system to study heterotypic contacts, and their differences with homotypic contacts.

*Drosophila* retina is composed of approximately 750 facets called ommatidia (*Cagan and Ready, 1989*; *Tepass and Harris, 2007*), each of which includes four cone cells (C) embedded in two primary pigment cells (P), along with other cell types shared by neighboring ommatidia (*Figure 1A,B*). The pattern of cone cells arrangement is strikingly similar to that of soap bubbles (*Hayashi and Carthew, 2004*). While this visual resemblance suggests that cells might minimize their surface of contact, both contractility and adhesion have to be considered for cell shape and cell arrangements (*Lecuit and Lenne, 2007*), as indicated by physical models (*Käfer et al., 2007*; *Hilgenfeldt et al., 2008*). Two classical Type I cadherins, E-cadherin (Ecad) and N-cadherin (Ncad) are expressed in the retina and specific expression of N-cadherin solely in cone cells governs the cone cell shape and arrangements (*Hayashi and Carthew, 2004*). In silico predictions based on energy minimization reproduce well the cone cell shapes but have limited experimental support (*Käfer et al., 2007*; *Hilgenfeldt et al., 2008*). In particular, the contributions of Ncad-mediated actomyosin contractility, as well as the interfacial tension in cone cell shape control, have not been explored.

Ncad is involved in numerous morphogenetic processes including cell migration, neural tube formation, and axon guidance (*Derycke and Bracke, 2004*; *Hirano and Takeichi, 2012*; *Lee et al., 2001*). To date, the direct implication of Ncad and actomyosin complexes on cell sorting and patterning during development is unclear. Ncad depletion in *Xenopus* neural plate leads to the loss of activated form of myosin light chain (*Nandadasa et al., 2009*). Actin cytoskeleton remodelling in *Drosophila* glial cells is tightly regulated by Ncad levels (*Kumar et al., 2015*). In cell culture, a dynamic interaction was reported between Ncad and actomyosin complexes in myocytes (*Comunale et al., 2007*; *Ladoux et al., 2010*; *Shih and Yamada, 2012*; *Chopra et al., 2011*), neurons (*Bard et al., 2008*; *Luccardini et al., 2013*; *Garcia et al., 2015*; *Okamura et al., 2004*; *Chazeau et al., 2015*) and fibroblasts (*Ouyang et al., 2013*).

Here we combine mechanical measurements, quantitative microscopy and modelling to revisit the role of Ncad in cell shapes and cell arrangement. We show that Ncad bonds contribute two fold less than Myosin-II (MyoII) to interfacial tension, but that Ncad also affects localization and levels of MyoII, and thus cell shapes. We reveal that heterotypic interfaces between Ncad-expressing and non-Ncad-expressing cells accumulate MyoII more than homotypic interfaces, thereby stabilizing specific cell configurations. Our results emphasize the interplay between cadherins and actomyosin networks, which determines cell shape and cell arrangements during morphogenesis.

## Results

### Cadherins and Myosin-II distribution in pupal retinas

To visualize the patterns of cadherins in ommatidia, we analyzed their expression in Ncad::GFP (*Figure 1C*) and Ecad::GFP knock-in retinas (*Figure 1D*) (*See Material and methods for details*). As previously reported (*Hayashi and Carthew, 2004*), Ncad is localized at cone cell-cone cell contacts (C|C), where it forms homophilic complexes (*Figure 1C*, *white arrowhead*). Ncad is also found at low level at the junctions between cone cell and primary pigment cell (C|P) (*Figure 1C*, *cyan arrowhead and Figure 1—figure supplement 1A*). At C|P contacts, Ncad cannot form trans-homophilic bonds but cis-homophilic bonds, as it is expressed in cone cells but not in primary pigment cells. In addition, Ncad-Ecad trans-heterotypic bonds appear to be absent, as *Ecad* mutant cone cell loses contact from the neighbouring Ecad expressing primary pigment cell (*Hayashi and Carthew, 2004*). Ecad is present in all cell contacts albeit at different levels. Ecad concentration is lower at C|C relative to C|P and at primary pigment cell and primary pigment cell contacts (P|P) (*Figure 1D*). To visualize the pattern of MyoII, we imaged Myosin heavy chain (Zip)::YFP knock-in retinas (*Figure 1E*), and Myosin light chain (Sqh)::GFP flies driven by *Sqh* promotor in *Sqh* mutant background (*Figure 1—figure supplement 1B*). We also stained Zip::YFP or Sqh::GFP retinas with Phospho-Myosin-II light chain antibodies which labels active MyoII to check how well they correlate with each other (*Figure 1—figure supplement 1C,D*). As reported earlier (*Warner and Longmore, 2009*;

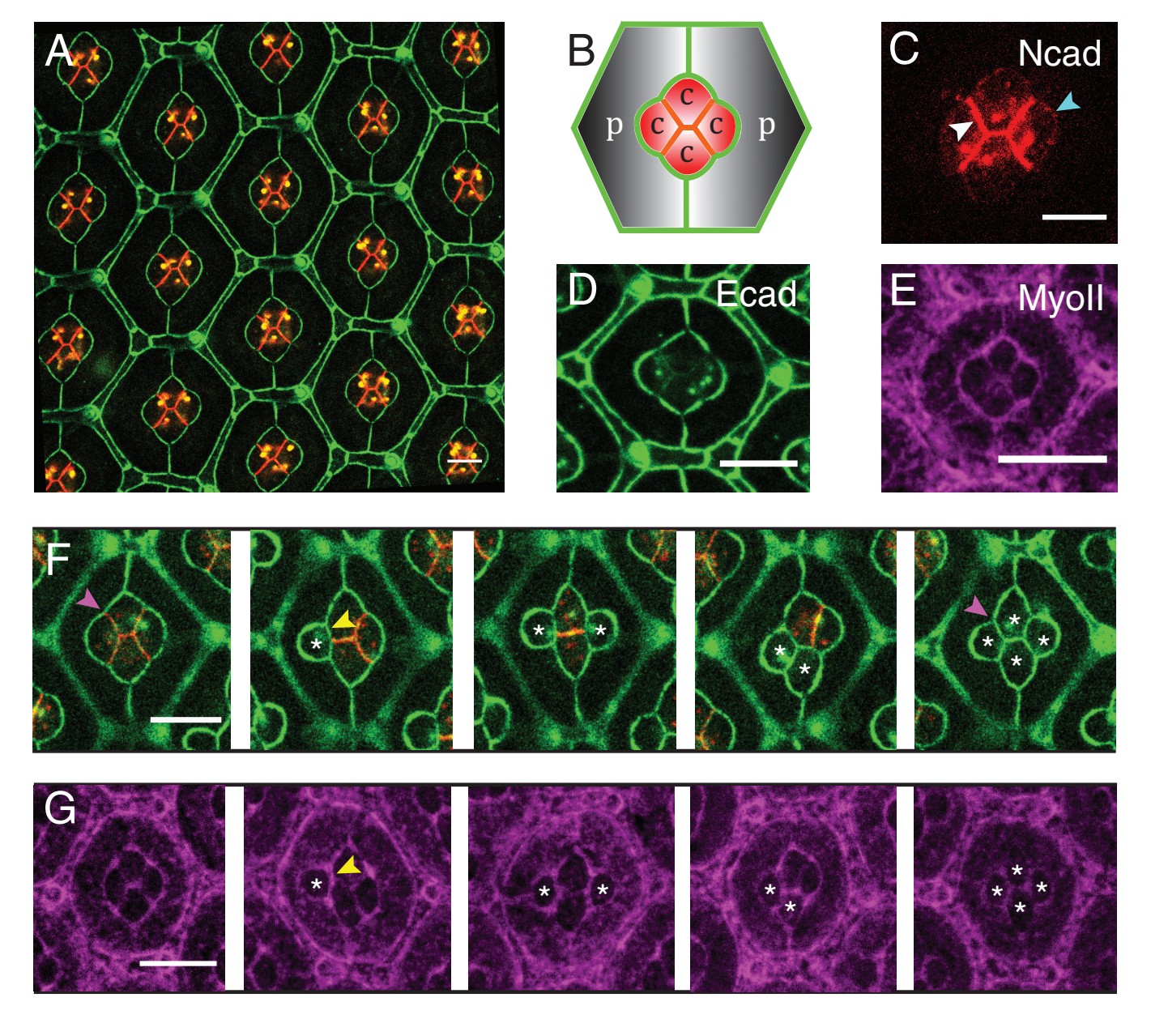

**Figure 1.** Patterns of *Drosophila* eye with the distributions of cadherins and Myosin-II (MyoII) in wildtype and *Ncad^M19* mosaic ommatidia. (**A**) Image of pupal retina at 41 hr after puparium formation (APF) consisting of repeating lattice structure called ommatidia labeled with Ecad::GFP (green) and Ncad::mKate2 (red). (**B**) A schematic of the most apical view of an ommatidium, which contains four cone cells (C) and two primary pigment cells (P), and the localization of cadherins (Ecad in green and Ncad in red). (**C–E**) An individual ommatidium with Ncad::GFP in red (**C**), Ecad::GFP in green (**D**), Zip::YFP in magenta (**E**). (**F–G**) Wildtype and *Ncad^M19* mosaic ommatidia labelled with Ecad::GFP (green), Ncad (red) and Zip::YFP (magenta). *Ncad^M19* cone cells are marked by white asterisks. Magenta arrowheads in (**F**) shows the angle change in full *Ncad^M19* cone cells compared to wildtype. White arrowhead indicates the C|C contact with homophilic complexes and cyan arrowhead indicates the C|P contact in (**C**). Yellow arrowheads indicate one of the contacts at the interface between wildtype and *Ncad^M19* cells to highlight the absence of Ncad adhesion in (**F**) and significant increase in MyoII levels in (**G**). Scale bar, 10 μm.

The following source data and figure supplements are available for figure 1:

**Figure supplement 1.** Ncad and MyoII levels using different reporters (Myosin-II light chain, Myosin-II heavy chain and Phospho-myosin-II light chain).

**Figure supplement 1—source data 1.** Dataset for *Figure 1- supplement figure 1A*.

*Yashiro et al., 2014*; *Deng et al., 2015*), Phospho-Myosin-II light chain antibodies show a punctate distribution, but overall the three markers indicate that MyoII is enriched at cell contacts and is also present as apical mesh at a lower concentration.

## Loss of N-cadherin not only affects cone cell shape but also Myosin-II levels

To assess the impact of Ncad on cone cell shapes, we performed mosaic analysis to generate Ncad loss of function (*Ncad^{M19}*) clones in pupal retinas. *Ncad* mutation in one or multiple cone cells results in significant cell shape changes (*Figure 1F*), as reported earlier (*Hayashi and Carthew, 2004*). Shape variations are dependent on the numbers and combinations of wildtype and *Ncad^{M19}* cone cells in the mosaic ommatidia (*Figure 1F*). In a full *Ncad^{M19}* ommatidium, the four cone cells acquire a cruciform shape rather than the normal diamond shape (last and first image respectively in *Figure 1F*). Reduction in cell contact length (*Figure 1F, yellow arrowhead*) and change in angles (*compare first and the last image of Figure 1F, magenta arrowhead*) suggests that adhesion by homophilic bonding of Ncad causes a significant expansion of contacts between cone cells. Apart from the cell shape changes, there are variations in MyoII levels at mosaic *Ncad^{M19}* ommatidium. For instance, at wildtype and *Ncad^{M19}* cone cell contact, there is a significant increase in MyoII level (*Figure 1G, yellow arrowhead*). So, the loss of *Ncad* induces change in MyoII concentrations, suggesting a possible contribution of MyoII contractility in shaping cone cell patterns (*Figure 1G*).

## Differential Myosin-II levels and interfacial tension

To explore the role of contractile forces in cone cell shapes, we determined the distribution of MyoII, a proxy for contractility, and measured interfacial tension acting at cell contacts in wildtype and *Ncad^{M19}* mosaic ommatidia.

We used Zip::YFP fluorescence intensity as a readout of MyoII concentration. We observed different levels of MyoII at cell contacts, depending on whether (i) the two cells, for example cell 1 and cell 2 in contact express both Ecad and Ncad (1(E,N)|2(E,N)), (ii) the two cells in contact express only Ecad (1(E)|2(E)), (iii) one of the two cells in contact expresses only Ecad and another expresses both Ecad and Ncad (1(E)|2(E,N)) (*Figure 1G*, yellow arrowhead and *Figure 1F,G*).

In wildtype ommatidia, MyoII level was found 2.2-fold higher at the contact between cone cell and primary pigment cell, C(E,N)|P(E), than at C(E,N)|C(E,N) contacts. MyoII at contacts between primary pigment cells, P(E)|P(E), was found 1.8-fold higher than at C(E,N)|C(E,N) contacts (*Figure 2A–C*, *Supplementary file 1* - table 1). A same trend in MyoII concentration is also observed when using Sqh::GFP as marker (*Figure 2—figure supplement 1A–C*, *Supplementary file 1* -table 1). Interestingly, in *Ncad^{M19}* mosaic ommatidium comprised of two *Ncad^{M19}* cone cells, we again observed three distinct levels of MyoII depending on the genotype of the two cone cells in contact (WT and WT (C(E,N)|C(E,N)), WT and *Ncad^{M19}* (C(E,N)|C(E)), *Ncad^{M19}* and *Ncad^{M19}* (C(E)|C(E))) (*Figure 2D–F* and *Figure 2—figure supplement 1D–H*, *Supplementary file 1* -table 1). These data revealed a simple gradation in MyoII concentration $c_{Myo}$, similar to the wildtype at C(E,N)|C(E,N), P(E)|P(E) and C(E,N)|P(E) contacts: $c_{Myo}(C(E,N)|C(E,N))=1$, $c_{Myo}(C(E)|C(E))=1.6$, and $c_{Myo}(C(E,N)|C(E))=2.3$ (in arbitrary units). Our data indicates that differences in MyoII concentrations at contact are dependent on Ncad expression in the cells.

Apart from MyoII, we also observed differences in Ecad levels when comparing C(E,N)|C(E,N), C(E)|C(E), C(E,N)|C(E) contacts (*Figure 2—figure supplement 1F,G,H*, *Supplementary file 1* -table 2), raising the possibility that the changes in MyoII levels might be a consequence of Ecad homotypic interactions. However, MyoII levels are uncorrelated with Ecad levels, ruling out this possibility (*Figure 2—figure supplement 1G and H*).

MyoII level anti-correlates with cell contact length (*Figure 2—figure supplement 1G,H*), which is consistent with the idea that MyoII regulates length. One can argue that the knowledge of MyoII distribution is not sufficient to characterize contractility and that F-actin distribution and organization might also be an important determinant (*Reymann et al., 2012*). Thus, we stained for F-actin using phalloidin and found that F-actin is mostly apical and junctional like MyoII, but its distribution does not strictly correlate with that of MyoII; homotypic C(E,N)|C(E,N) contacts show higher F-actin level than C(E,N)|P(E), C(E,N)|C(E), P(E)|P(E) and C(E)|C(E) contacts (*Figure 2—figure supplement 2A–C*).

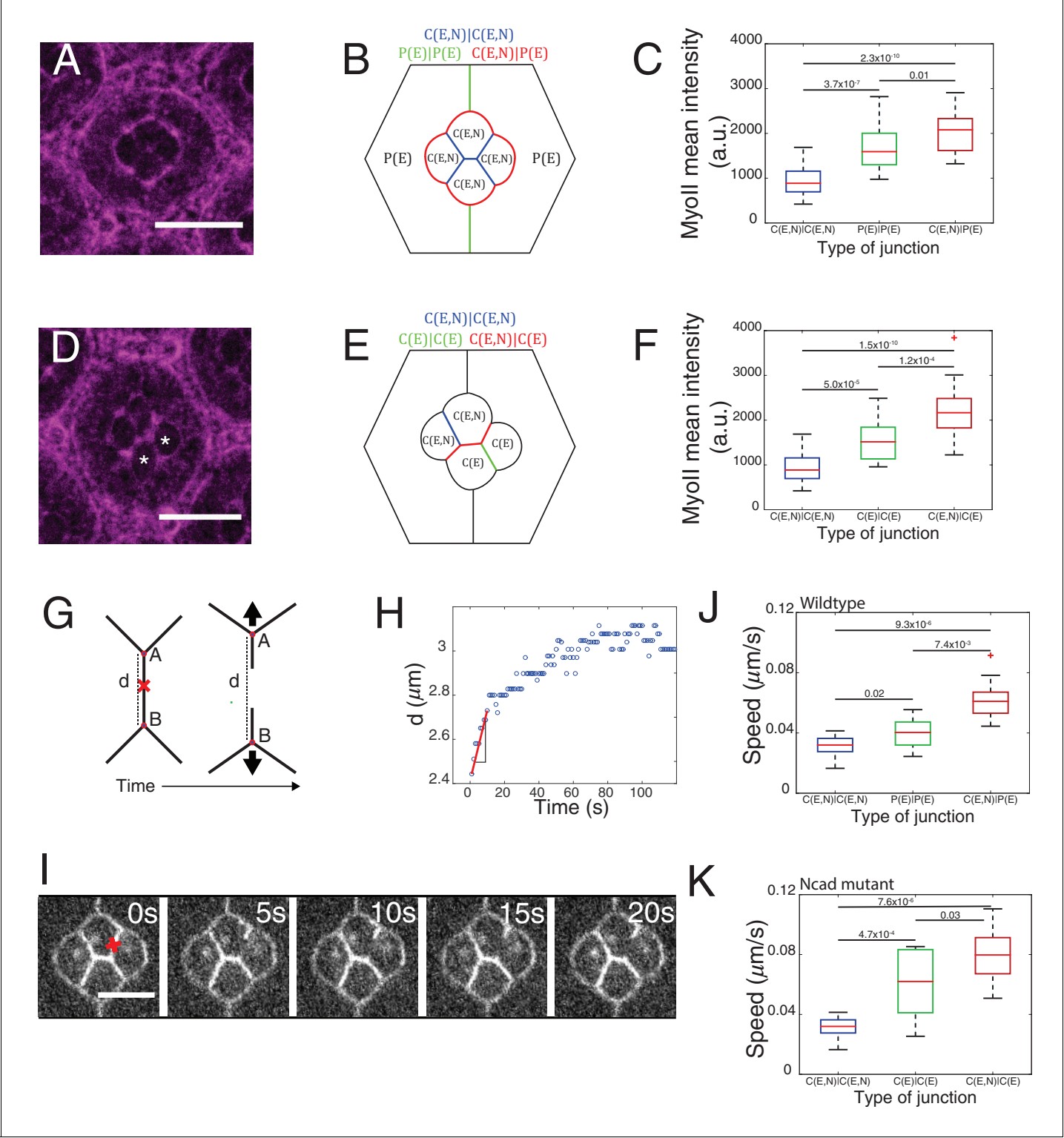

**Figure 2.** Differential MyoII levels and interfacial tensions at various cell contacts. (**A**) Wildtype ommatidium with Zip::YFP represented by (**B**) a schematic that highlights three different types of contacts at cell interfaces that express Ecad or Ncad or both E and Ncad. C(E,N)|C(E,N) contact (blue) shared by two cone cells, P(E)|P(E) contact (green) shared by two primary pigment cells and C(E,N)|P(E) contact (red) shared by a cone and a primary pigment cell. Scale bar, 10 μm. (**C**) Quantification of MyoII intensity in C(E,N)|C(E,N) (n = 30), P(E)|P(E) (n = 22) and C(E,N)|P(E) (n = 36) contacts. P-values are shown above the black horizontal lines (non-parametric Mann-Whitney U test on pairs and Bonferroni correction). (**D**) A *Ncad*[M19] mosaic ommatidium with Zip::YFP. *Ncad*[M19] cells are marked by white asterisks. Scale bar, 10 μm. (**E**) A schematic represents the corresponding *Ncad*[M19]

*Figure 2 continued on next page*

*Figure 2 continued*

mosaic mutants highlighting C(E,N)|C(E,N) (blue), C(E)|C(E) (green) and C(E,N)|C(E) contacts (red). (F) Quantification of MyoII intensity in C(E,N)|C(E,N) (n = 30), C(E)|C(E) (n = 22) and C(E,N)|C(E) (n = 36) contacts. P-values are shown above the black horizontal lines. (G)-(K) Laser nanoablation experiments to estimate interfacial tension. (G) Schematic of a contact before (left) and after (right) ablation. Red cross represents the point of the ablation. Vertex A and B recoil changing distance 'd' after ablation. (H) Opening curve plots the distance' d' over time with a linear fit for initial time points to get the initial recoil speed. (I) Snapshot of an ablation at C(E,N)|C(E,N) contact in wildtype ommatidium, red cross indicates the ablation point. (J) Quantification of initial recoil speed of C(E,N)|C(E,N) (n = 14), P(E)|P(E) (n = 18) and C(E,N)|P(E) (n = 19) contacts in wildtype ommatidia. P-values are shown above the black horizontal lines. (K) Quantification of initial recoil speed in C(E,N)|C(E,N) (n = 14), C(E)|C(E) (n = 18) and C(E,N)|C(E) (n = 17) contacts in *Ncad$^{M19}$* mosaic mutants. Scale bar, 5 μm. P-values are shown above the black horizontal lines.

The following source data and figure supplements are available for figure 2:

**Source data 1.** Dataset for *Figure 2C,F,J and K*.

**Figure supplement 1.** Junction length, cadherins and MyoII levels at different contacts.

**Figure supplement 1—source data 1.** Dataset for *Figure 2—figure supplement 1C,G and H*.

**Figure supplement 2.** *Ncad$^{M19}$* mosaic ommatidium interfacial tension measurements and F-actin distribution.

**Figure supplement 2—source data 1.** Dataset for *Figure 2—figure supplement 2B and C*.

In an attempt to determine the relationship between MyoII-dependent contractility and tensile forces at cell contacts, we performed laser nano-dissection experiments (*Rauzi et al., 2008*). The initial recoil speed after the cell contact ablation served as a proxy for interfacial tension (*Figure 2G–I*, *Figure 2—figure supplement 2D*, *Videos 1* and *2*). We found that tension at C(E,N)|C(E) contacts was the highest while tension at C(E,N)|C(E,N) contacts was the lowest (*Figure 2J,K*). These values correlate with the levels of MyoII (compare *Figure 2C and J* or *Figure 2F and K*) and are consistent with the hypothesis that MyoII is a major determinant of interfacial tension.

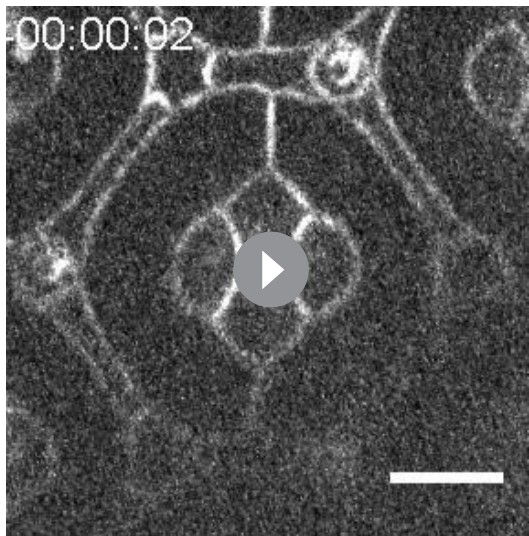

**Video 1.** Laser nano-ablation of C(E,N)|C(E,N) contact in wildtype ommatidium. Ablation at 00:00:00. Frame rate is 1 s/frame. Labelling: β-cat::GFP. Scale bar, 5 μm.

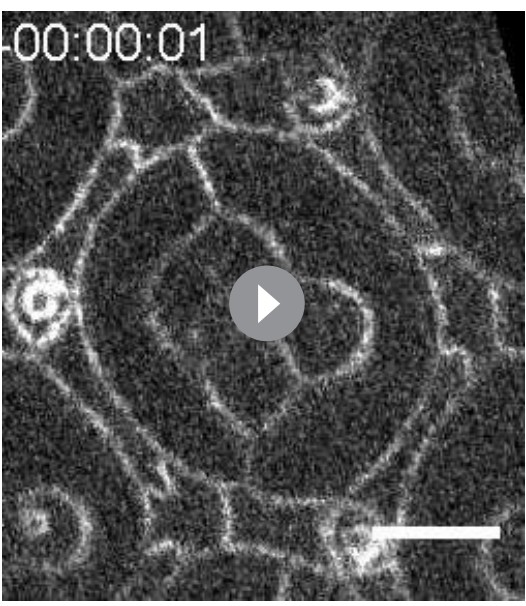

**Video 2.** Laser nano-ablation of C(E,N)|C(E) contact in *Ncad$^{M19}$* mosaic ommatidium with polar (Pl) and posterior (P) cone cells(see *Figure 5A* for cone cell axes of polarity) lacking *Ncad*. Ablation at 00:00:00. Frame rate is 250 ms/frame. Labelling: Ecad::GFP. Scale bar, 5 μm.

## Bound and unbound N-cadherin differentially impact on Myosin–II junctional localization

To determine whether and how Ncad might control cell shape through MyoII regulation, we focused on the links between Ncad and MyoII localization. We observed high level of Ncad at homotypic contacts (C(E,N)|C(E,N)) which also exhibit the lowest concentration of MyoII, by 1.8 fold lower than the P(E)|P(E) cell contacts. This suggests that homophilic Ncad at homotypic contact reduces MyoII levels (*Figure 2A,C*), in agreement with the idea that cadherin lowers interfacial tension at cell contacts (*Maître and Heisenberg, 2013*). At heterotypic contacts (C(E,N)|P(E)), where Ncad cannot form transhomophilic bond, Ncad was found at very low level (*Figure 1C*, *Figure 1—figure supplement 1A*) and MyoII at a higher level than at any other contact (*Figure 2A,C*). This suggests that unbound Ncad at heterotypic contact signals to MyoII and induces its accumulation. To confirm this hypothesis, we took advantage of the fact that the primary pigment cells do not express Ncad and asked if we could modify MyoII level at different cell contacts by Ncad misexpression.

Ncad misexpression in one of the primary pigment cell affected the shape of cone cells in contact with it (*Figure 3A,B*). Homophilic Ncad was detected at the C(E,N)|P(E,N+) contacts (*Figure 3—figure supplement 1A and A', yellow arrowhead*) and MyoII levels at these modified contacts (C(E,N)|P(E,N+), *Figure 3A,B, yellow arrowhead*) were significantly reduced compared to wildtype C|P contacts (C(E,N)|P(E), *Figure 3A,B, green arrowhead, Supplementary file 1* - table 1). This confirms our hypothesis that homophilic Ncad reduces MyoII level (*Figure 3C*). In addition, higher level of MyoII was detected at contacts between primary pigment cells with one of them misexpressing Ncad (P(E)|P(E,N+)) (*Figure 3A,D red arrowhead*) than at contacts between wildtype primary pigment cells expressing only Ecad (P(E)|P(E)) (*Figure 3A,B,D, Supplementary file 1* -table 1).

To test whether such property of Ncad is specific to the retinal epithelium or more general, we performed clonal misexpression of Ncad in the larval wing pouch which cells express only Ecad. We noticed higher level of MyoII at the boundary of clones compared to MyoII inside the clones or to the surrounding wildtype tissue (*Figure 3—figure supplement 1B,C, cyan arrowheads*). This indicates that MyoII regulation by Ncad is not specific to the retina.

At C(E,N)|P(E) contacts, Ncad is asymmetrically localized as it is expressed only in one of the two apposed cells. We thus wondered whether MyoII could also be asymmetrically localized. To address this, we measured the intensity profile of MyoII perpendicular to C(E,N)|P(E) contacts, using Ecad intensity as a marker for the contact position. Localization of Ecad::GFP, and thus the contact position, can be determined with a precision better than the diffraction limit given the high signal/noise ratio (5–22 nm) (*Figure 3—figure supplement 2, See Materials and methods*). We found that MyoII maximum intensity was systematically shifted towards the cell that expressed both Ecad and Ncad (*Figure 3—figure supplement 2A,B*). Importantly, the distance between MyoII and Ecad intensity peaks (*Figure 3—figure supplement 2C*) was found larger than the imprecision in peaks' localization (*Figure 3—figure supplement 2D, See Material and methods*). This significant and systematic shift indicates that MyoII is enriched in the cortex of an Ecad- and Ncad-expressing cell when it is apposed to an Ecad-expressing cell (*Figure 3E–G*). Using Starry night (Stan), a membrane marker that has a higher signal/noise ratio than Ecad at C(E,N)|C(E,N) contacts (*Figure 3—figure supplement 2E*), we confirmed the asymmetry of MyoII at C(E,N)|P(E) contacts (*Figure 3—figure supplement 2F,G*). In contrast, we observed a symmetric localization of MyoII at C(E,N)|C(E,N) contacts (*Figure 3—figure supplement 2F,H*).

This increase in MyoII level is cell contact autonomous: we observed higher MyoII intensity at C(E,N)|P(E) contacts, irrespective of the other contacts of the cell (for instance, C(E,N)|C(E,N)). This increase is striking in $Ncad^{M19}$ mosaic ommatidia in which a single Ecad- and Ncad-expressing cell is surrounded by Ecad-expressing cells: we noticed an intense ring of MyoII at the cortex (*Figure 3—figure supplement 3A,B,C, compare cells marked by white and green arrowheads, Figure 3—figure supplement 3C*). To further confirm the above observation, Ncad was again misexpressed in primary pigment cells to check if it could induce MyoII asymmetry at the modified P(E)|P(E,N+) contacts. An asymmetric localization of MyoII in cells that express both Ecad and Ncad was observed at the P(E)|P(E,N+) contacts (*Figure 3H–J*).

To further explore how Ncad at heterotypic contacts could induce MyoII contractility, we expressed only the extracellular part of Ncad in one primary pigment cell (*Figure 4A, white +*). Such truncated Ncad can form adhesion bonds but cannot interact with the actomyosin network

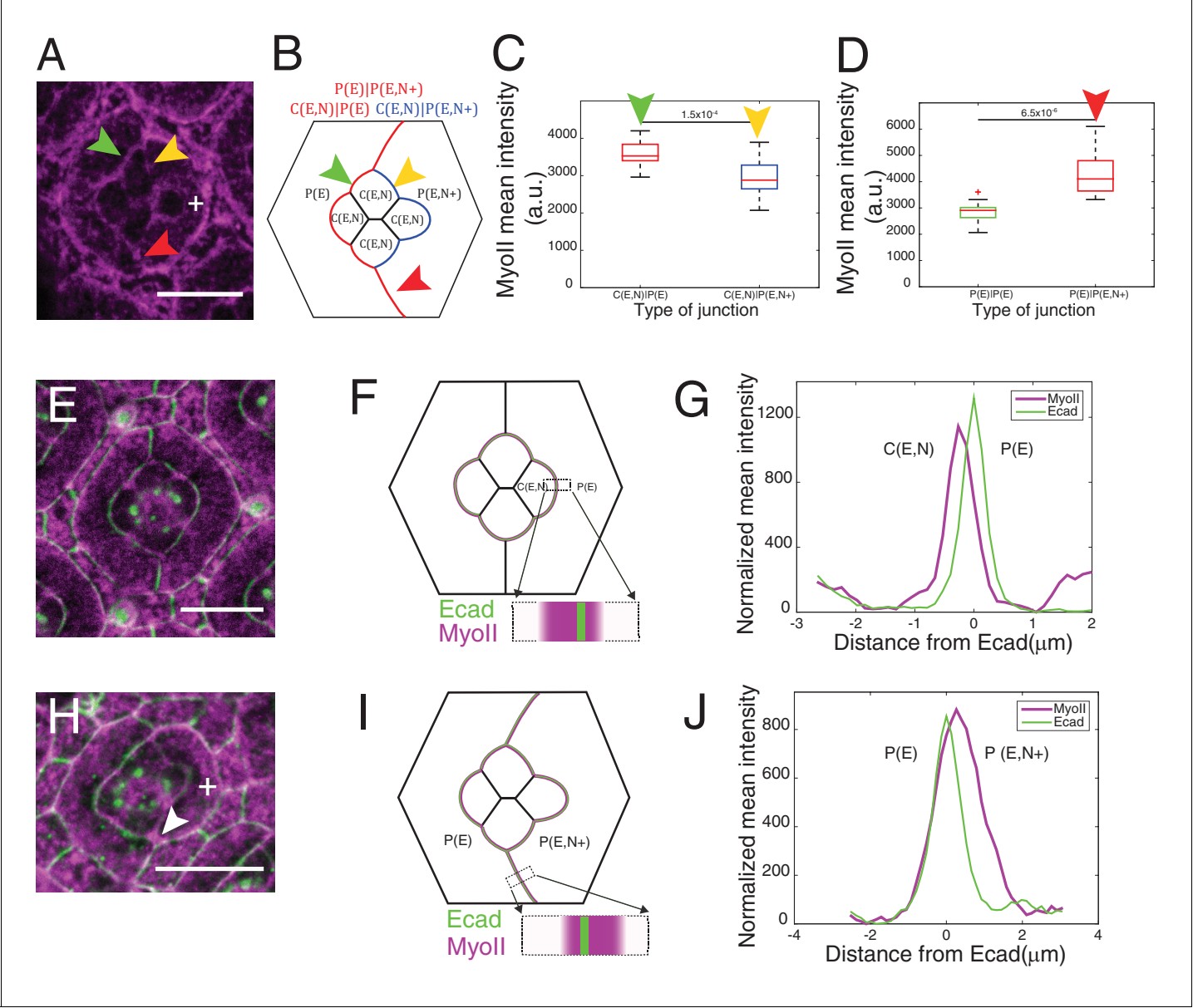

**Figure 3.** Misexpression of Ncad in primary pigment cells and MyoII accumulation and MyoII asymmetry at cell contacts. (**A**) An ommatidium with Ncad misexpressed in one of the primary pigment cells (white **+**) with Zip::YFP in magenta. Green arrowhead indicates the C(E,N)|P(E) contact. Yellow and red arrowheads indicate the modified C(E,N)|P(E,N+) and P(E)|P(E,N+) contacts respectively. (**B**) A schematic of Ncad misexpression ommatidium with the modified C(E,N)|P(E,N+) (blue), wildtype C(E,N)|P(E) and modified P(E)|P(E,N+) (red) contacts. (**C**) Quantification of MyoII intensity in C(E,N)|P(E) (n = 20) and C(E,N)|P(E,N+) (n = 20) contacts. P-value is shown above the black horizontal line. (**D**) Quantification of MyoII intensity in P(E)|P(E) (n = 16) and P(E)|P (E,N+) (n = 16) contacts. P-value is shown above the black horizontal line. (**E**) Wildtype ommatidium with Ecad::GFP (green) and Sqh::Ch (magenta). (**F**) Schematic with a zoom-in of a C(E,N)|P(E) contact shared by cone cell and primary pigment cell representing the asymmetric distribution of MyoII. (**G**) Average linescan of Sqh::Ch (magenta) intensity with respect to Ecad::GFP intensity (green) normal to interfaces (n = 10 interfaces). (**H**) An ommatidium with Ncad misexpressed in one of the primary pigment cell (white **+**) with Sqh::Ch (magenta). White arrowhead indicates the modified P(E)|P(E,N+) contact. (**I**) Schematics with a zoom-in of a modified P(E)|P(E,N+) contact shared by primary pigment cell and Ncad misexpressed primary pigment cell representing the asymmetric distribution of MyoII. (**J**) Average linescan of Sqh::Ch intensity (magenta) with respect to Ecad::GFP intensity (green) (n = 13 interfaces). Scale bar 10 μm.

The following source data and figure supplements are available for figure 3:

**Source data 1.** Dataset for *Figure 3C,D,G and H*.

**Figure supplement 1.** Misexpression of Ncad in primary pigment cell in retinas and larval wing pouch.

*Figure 3 continued on next page*

*Figure 3 continued*

**Figure supplement 2.** Asymmetry of MyoII localization at different contacts.

**Figure supplement 2—source data 1.** Dataset for *Figure 3—figure supplement 2D,F,G and J*.

**Figure supplement 3.** MyoII levels of a single wildtype cone cell in *Ncad^M19* mosaic ommatidium.

**Figure supplement 3—source data 1.** Dataset for *Figure 3—figure supplement 3C*.

(*Figure 4A, white arrowhead*). We observed a change in contact shape and MyoII levels at the interface between the wildtype cone cell and primary pigment cell that misexpressed extracellular Ncad (*Figure 4B–D, compare blue and red arrowheads, Supplementary file 1* -table 1), which confirms a role for homophilic Ncad bonds in the downregulation of MyoII contractility. However, MyoII levels at the contact between primary pigment cells, which included one cell that misexpressed extracellular Ncad showed no change in MyoII, when compared to full-length Ncad (*Figure 4B,C,E, green arrowhead, Supplementary file 1* -table 1). This result suggested that cytoplasmic part of Ncad is required for the accumulation of MyoII at the C(E,N)|P(E) contacts.

The above data suggest that while homophilic Ncad reduces MyoII contractility at homotypic contacts, unbound Ncad is able to activate MyoII, and locally enhance contractility at heterotypic contacts through its cytoplasmic part.

## Both local tension and cell-scale contractility determine ommatidia shape

To understand how tensions at cell contacts determine ommatidia shape, we sought to build a simple mechanical model integrating both local tension and cell-scale contractility. Following earlier works, we thus designed a 2D model based on the minimization of a tension-based energy function (*Käfer et al., 2007*; *Hilgenfeldt et al., 2008*; *Farhadifar et al., 2007*). Although retina is obviously a 3D system, we treat the plane of adherens junctions, where both adhesion and MyoII molecules are recruited, as a 2D system. Since retinal cells have a complex shape and are variant in the z-direction, the relevance of the model is therefore limited to the junctional plane. Such an energy-based model assumes that the system settles to a configuration of minimum potential energy, which is likely to be the case in vivo since the developmental process is very slow and quasi-static. We then assume that individual contacts have a local tension $\gamma_{loc}$. As shown by our experiments, $\gamma_{loc}$ is likely to be determined by the concentration of MyoII and cadherins engaged at the contact. The contribution of $\gamma_{loc}$ at each contact to the total energy of the system is simply $\gamma_{loc}l$, where $l$ is the contact length. In addition, and as shown by others (*Käfer et al., 2007*; *Hilgenfeldt et al., 2008*; *Farhadifar et al., 2007*), the contractile cortical network and the 3D cell volume constraint are likely to impose a 2D geometry constraint at the cell level. We encapsulate this in a perimeter elasticity term, in which deviations $\Delta p$ of the cell perimeter $p$ from a preferred cell perimeter $p_0$ yield an energy penalty $\frac{K}{2}\frac{(p-p_0)^2}{p_0}$. The elastic constant $K$, which we assume is the same for all cells, determines how big this penalty is. In two-dimension, the mechanical energy of the ommatidium thus writes:

$$E = \sum_{\text{contact } ij} \gamma_{locij}l_{ij} + \sum_{\text{cells } i} \frac{K}{2}\frac{(p_i - p_{0i})^2}{p_{0i}}$$

While cell area can vary experimentally, in a range which is likely to be determined by volume constraint and cell elasticity, in the model we chose to fix the area using a Lagrange multiplier. This choice is driven by simplicity arguments. Unlike perimeter elasticity, area elasticity is not crucial to select a shape or configuration, but mostly set the cell size (*Hilgenfeldt et al., 2008*). Interfacial tension at a cell junction is, by definition, the derivative of the energy function with respect to junction length, and writes:

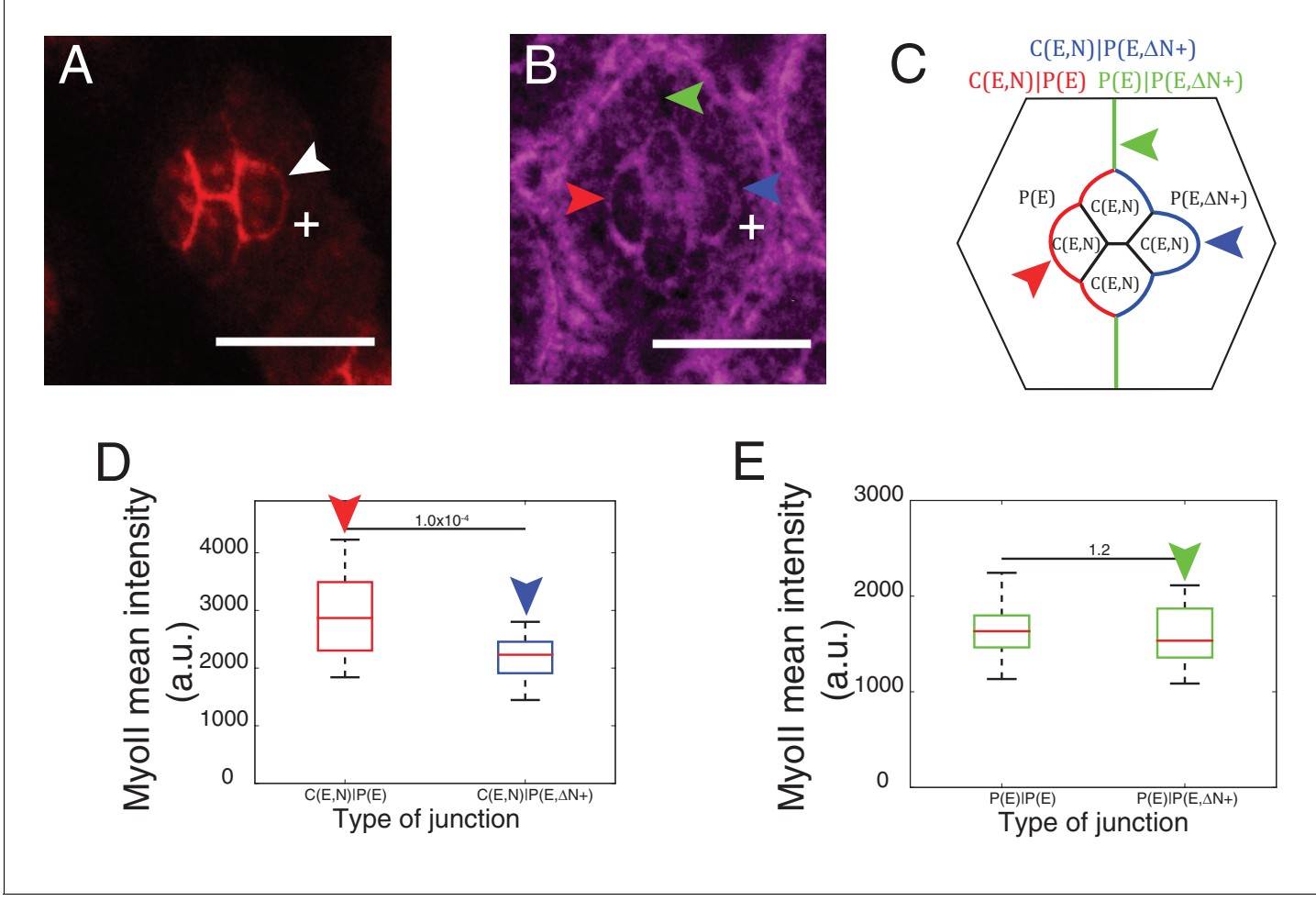

**Figure 4.** Cytoplasmic part of Ncad is required for MyoII accumulation in heterotypic contacts. (A–B) An ommatidium misexpressing extracellular part of Ncad in one of the primary pigment cells (white +) with Ncad (A) and Zip::YFP (B). White arrowhead indicates the C(E,N)|P(E,ΔN+) cell contact with homophilic Ncad in (A), red arrowhead indicates C(E,N)|P(E,ΔN+) wildtype cell contact, blue arrowhead indicates modified C(E,N)|P(E) cell contact and green arrowhead indicates the unchanged P(E)|P(E,ΔN+) cell contact. (C) Schematic of ommatidium misexpressing extracellular part of Ncad shows the modified cell contacts, C(E,N)|P(E) contact (blue), wildtype C(E,N)|P(E) contact (red) and unaffected P(E)|P(E,ΔN+) contact (green). (D) Quantification of MyoII intensity in C(E,N)|P(E) (n = 28) and C(E,N)|P(E,ΔN+) (n = 28). P-value is shown above the black horizontal line. (E) Quantification of MyoII intensity in wildtype P(E)|P(E) (n = 19) and unaffected P(E)|P(E,ΔN+) contact (n = 19). Scale bar, 10 μm. P-value is shown above the black horizontal line.

The following source data is available for figure 4:

**Source data 1.** Dataset for *Figure 4D and E*.

$$\gamma_{ij} = \gamma_{locij} + K\frac{\Delta p_i}{p_{0i}} + K\frac{\Delta p_j}{p_{0j}} \tag{1}$$

Interfacial tension $\gamma$ is thus the sum of the local term, $\gamma_{loc}$, and of a cell-scale elastic term, $\gamma_{el} = K\frac{\Delta p_i}{p_{0i}} + K\frac{\Delta p_j}{p_{0j}}$. Note that ablation experiments reveal the global interfacial tension $\gamma$.

The parameters of the model are the target perimeters, the local tensions, and $K$. We sought to determine as many parameters as possible from experiments. We reasoned that in the absence of forces applied by surrounding cells, cells should acquire their preferred (target) perimeter (*Figure 5—figure supplement 1A*). We thus performed circular ablations, separating a cell from all its neighboring cells to measure the target perimeter. After ablation, cells relaxed towards a circular shape in the plane of adherens junctions (*Video 3*). Note that the perimeter after relaxation was

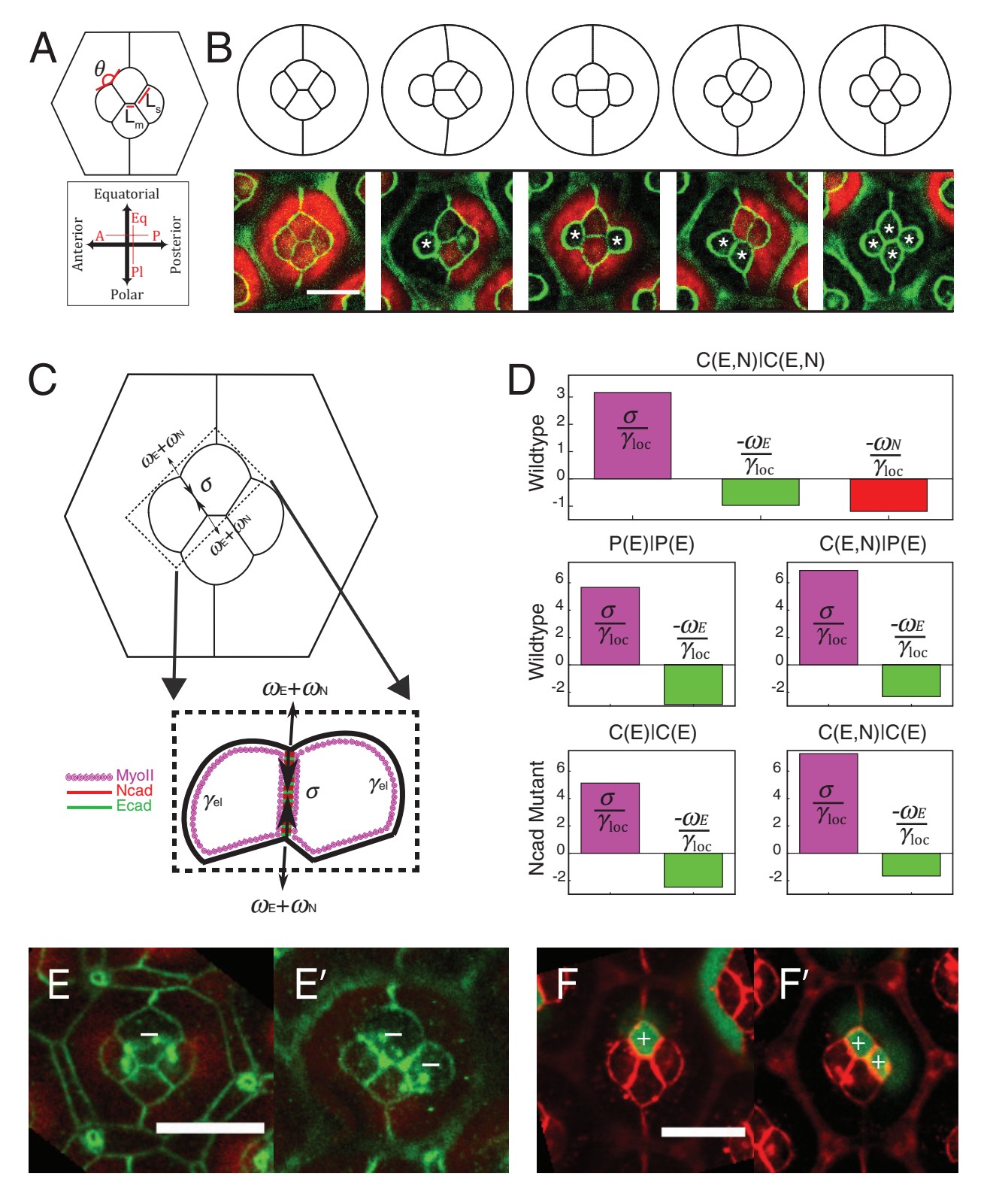

**Figure 5.** Simulations of cone cell shapes and contribution of cadherins and MyoII to cell shapes. (A) Schematics of two axes of polarity, A-P and Eq-Pl, of cone cells (bottom) and fit parameters measured in experiments and simulations, contact angle between cone cell and primary pigment cell (θ), ratio of contact length shared by A/P and Eq/Pl cell ($L_s$) to contact length shared by Eq and Pl cells ($L_m$) (top). (B) Comparison of experimental images (lower panel) to the simulations (upper panel), $Ncad^{M19}$ cells are marked by white asterisks. (C) Schematic of force balance resulting from adhesion of Ecad

*Figure 5 continued on next page*

*Figure 5 continued*

($\omega_E$, green) and Ncad ($\omega_N$, red), MyoII dependent cortical tension at the cell contact ($\sigma$) and cortex elasticity due to actomyosin at the cell perimeter ($\gamma_{el}$) (both in magenta). (**D**) Relative contribution of MyoII dependent cortical tension ($\sigma$), Ecad adhesion ($w_E$) and Ncad adhesion ($w_N$) to the local tension term $\gamma_{loc}$ for all contact types in wildtype and $Ncad^{M19}$ mosaic mutants. (**E–E'**) Image of the ommatidium with (**E**) Eq and (**E'**) Eq and Pl cone cells $Sqh^{Ax3}$ mutant (white -). $\beta$-catenin staining in green. (**F–F'**) Image of the ommatidium with (**F**) Eq and (**F'**) Eq and Pl cone cells expressing constitutively active form of Sqh, $Sqh^{T20ES20E}$ (white +), $\beta$-catenin staining in red. Scale bar, 10 µm.

The following source data and figure supplements are available for figure 5:

**Figure supplement 1.** Perimeter elasticity and determination of elastic constant (*K*).

**Figure supplement 1—source data 1.** Dataset for *Figure 5—figure supplement 1C*

**Figure supplement 2.** Elastic and local tension contribution to interfacial tension and comparison of simulation to experiment.

**Figure supplement 2—source data 1.** Dataset for *Figure 5—figure supplement 2C and D*.

**Figure supplement 3.** Ecad intensity measurements and correlation of interfacial tension to molecular distributions.

**Figure supplement 3—source data 1.** Dataset for *Figure 5—figure supplement 3B and F*.

**Figure supplement 4.** MyoII perturbations and simulations.

**Figure supplement 4—source data 1.** Dataset for *Figure 5—figure supplement 4A,B and E*.

found to be typically 8% smaller (8.4 ± 1.2, n = 7) than prior to ablation (*Figure 5—figure supplement 1B,C*). In addition, laser ablation experiments in *Figure 2* provided us with relative estimates of the interfacial tensions ($\gamma$) for the different contact types (C(E,N)|C(E,N), P(E)|P(E) and C(E,N)|P(E)). Note that all tensions (including *K*, which has the dimension of $\gamma$) were normalized by the interfacial tension measured for C(E,N)|C(E,N) contacts, and therefore are given in units of C(E,N)|C(E,N)=1. Using that $\gamma_{loc} \approx \gamma - 2K\frac{\Delta p}{p_o}$ and having determined $\frac{\Delta p}{p_o}$, *K* is the only free parameter remaining in the model. To determine its value, we minimized the energy function using the Surface Evolver software starting from an unrealistic configuration (*Figure 5—figure supplement 2A*), until the equilibrium configuration was reached. We then fitted the resulting ommatidia shapes to experimental shapes using *K* as a fit parameter. To fit simulations to experimental geometries, we chose two geometrical descriptors: the angle formed by adjacent C|P contacts and the length ratio between two contacts (polar-equatorial (*Lm*) over polar-posterior (*Ls*) contacts) (*Figure 5A*). We simulated the wildtype and four different $Ncad^{M19}$ mosaic ommatidia, and applied a weighted least squares method to fit them altogether (*Figure 5—figure supplement 2B*). The best fit corresponds to $K = 4.2$ (in units of $\gamma_{C(E,N)|C(E,N)}=1$). The cell patterns obtained in silico for this value are in very good agreement with the cell patterns observed in vivo, for wildtype ommatidia and for $Ncad^{M19}$ mosaic ommatidia with different numbers and combinations of wildtype and $Ncad^{M19}$ cone cells (*Figure 5B*, *Figure 5—figure supplement 2C,D*). Interestingly, our estimate of *K* also indicates that elastic tension contributes to 1/3 to 1/2 of the total interfacial tension, depending on the cell contacts considered (*Figure 5—figure supplement 2E*).

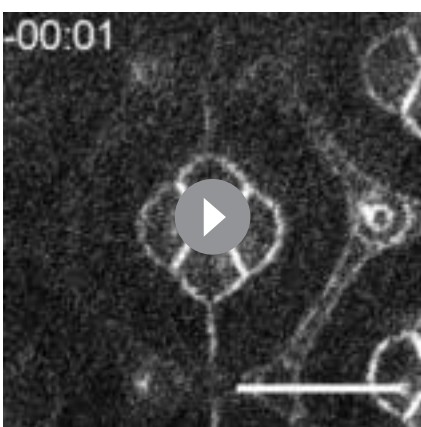

**Video 3.** Laser nano-ablation of target perimeter measurement ($\Delta p/p_o$). Ablation at 00:00. Frame rate is 250 ms/frame. Labelling: $\beta$-cat::GFP. Scale bar, 5 µm.

## The balance of cortical tension and adhesion determines local tension

The rationale of the model presented above is to predict ommatidia shapes from tensions at the cell contacts measured by ablations, irrespective of MyoII or cadherin levels. Yet, local tension is likely to result from the balance between MyoII-dependent cortical tension and cadherin-based adhesion (*Lecuit and Lenne, 2007*), and we were interested in weighing their respective (direct) contributions. To do so, we measured concentrations of cadherin and MyoII molecules in different configurations for which we knew the local tension.

We assumed that adhesion molecules and motor molecules have an additive and antagonistic contribution to local tension (*Maître et al., 2012*). Hence, MyoII cortical tension $\sigma$ is balanced by cadherin based adhesion $\omega$, so that $\gamma_{loc} = \sigma - \omega$ (*Figure 5C*). At C(E,N)|C(E,N) contacts, both Ecad and Ncad contribute to the adhesion term, so that $\omega = \omega_E + \omega_N$, while at C(E,N)|P(E) and P(E)|P(E) contacts, only Ecad contributes to adhesion, and $\omega = \omega_E$. For the sake of simplicity, we assumed that adhesion and MyoII-dependent cortical tension were proportional to the concentrations of cadherins and MyoII, respectively. It should be noted that whether MyoII molecules are recruited through an Ncad feedback or any other pathway is not relevant to how they contribute to local tension. Hence the feedback between MyoII and Ncad is not considered to estimate the respective contribution of cadherin and MyoII molecules to tension. From there, we could use the molecular concentrations (*Figure 2C,F*, *Figure 5—figure supplement 3A,B and E,F*) and local tensions $\gamma_{loc}$ obtained from ablation experiments combined to numerical modelling (*Figure 2J,K* and *Figure 5—figure supplement 2E*) to infer the contributions of Ecad, Ncad and MyoII to the local tension of the different contact types (See Materials and methods). We found that MyoII has a very significant contribution to local tension, which is about two to five times higher than that of Ncad or Ecad depending on the contact type (*Figure 5D*). This data, in agreement with in vitro experiments on cell doublets (*Maître et al., 2012*), emphasizes the quantitative role of MyoII on cell shapes in vivo. It also indicates that control of cell shape by adhesion is mostly indirect, through the regulation of MyoII level by unbound Ncad. This is again exemplified by the higher contribution of MyoII to local tension in C(E,N)|P(E) and C(E,N)|C(E) contacts than in P(E)|P(E) and C(E)|C(E) contacts (*Figure 5D*, *middle and bottom panels*).

To confirm the importance of MyoII on cone cell shapes, we manipulated MyoII activity in cone cells. We first decreased MyoII contractility using Myosin-II light chain loss of function ($Sqh^{Ax3}$) mutant (*Figure 5E,E'*), and observed a massive increase in cell apical area in the mutant cells and change in cell contact length (*Figure 5—figure supplement 4A Supplementary file 1* - table 2). Conversely, misexpression of constitutively active form of MyoII ($Sqh^{T20E.S21E}$) leads to a reduction of cell apical area (*Figure 5F,F'*) and change in cell contact length (*Figure 5—figure supplement 4B*). These changes in apical area suggest that shape changes resulting from MyoII loss of function or misexpression are dominated by cell-scale (apical MyoII) rather than cell contact-scale contribution of MyoII. This is exemplified by our simulations, in which a simple change of the (fixed) area yields a qualitatively good prediction of cone cell shapes in different mutant configurations (*Figure 5—figure supplement 4*, *See Materials and methods*). A more quantitative assessment on these experiments would most likely require additional terms of area elasticity. Note that to exemplify experimentally the contribution of MyoII to local tension, one would ideally want to selectively downregulate or upregulate MyoII at cell contacts only, which is technically very challenging.

## Myosin-II localization mediated by N-cadherin regulates cell arrangement

Lastly, to test the relevance of our data for tissue morphogenesis, we analyzed ommatidia morphogenesis in wildtype and $Ncad^{M19}$ mosaic retinas, 21 hr after pupal formation (APF) for 5 and 9 hr, respectively (*Figure 6A,B* and *Videos 4* and *5*). Wildtype cone cells undergo stereotypic neighbor exchanges (*Figure 6A*). Anterior and posterior cells lose A-P contact, while equatorial and polar cells intercalate and form a new Eq-Pl contact (A-P to Eq-Pl contact transition) (*Figure 5A*). However, when imaging the $Ncad^{M19}$ mosaic mutants, we observed defects of this A-P to Eq-Pl transition. 98,2% of analyzed ommatidia where $Ncad$ was mutated in either the equatorial or polar cell failed to transit (*Figure 6B*, *red arrowheads*, *Figure 6C*, n = 114). 100% of analyzed ommatidia where $Ncad$ was mutated in both equatorial and polar cells failed to transit (*Figure 6D*, n = 16). We reasoned that this transition might be prevented due to the increase in tension at the transverse cell contacts

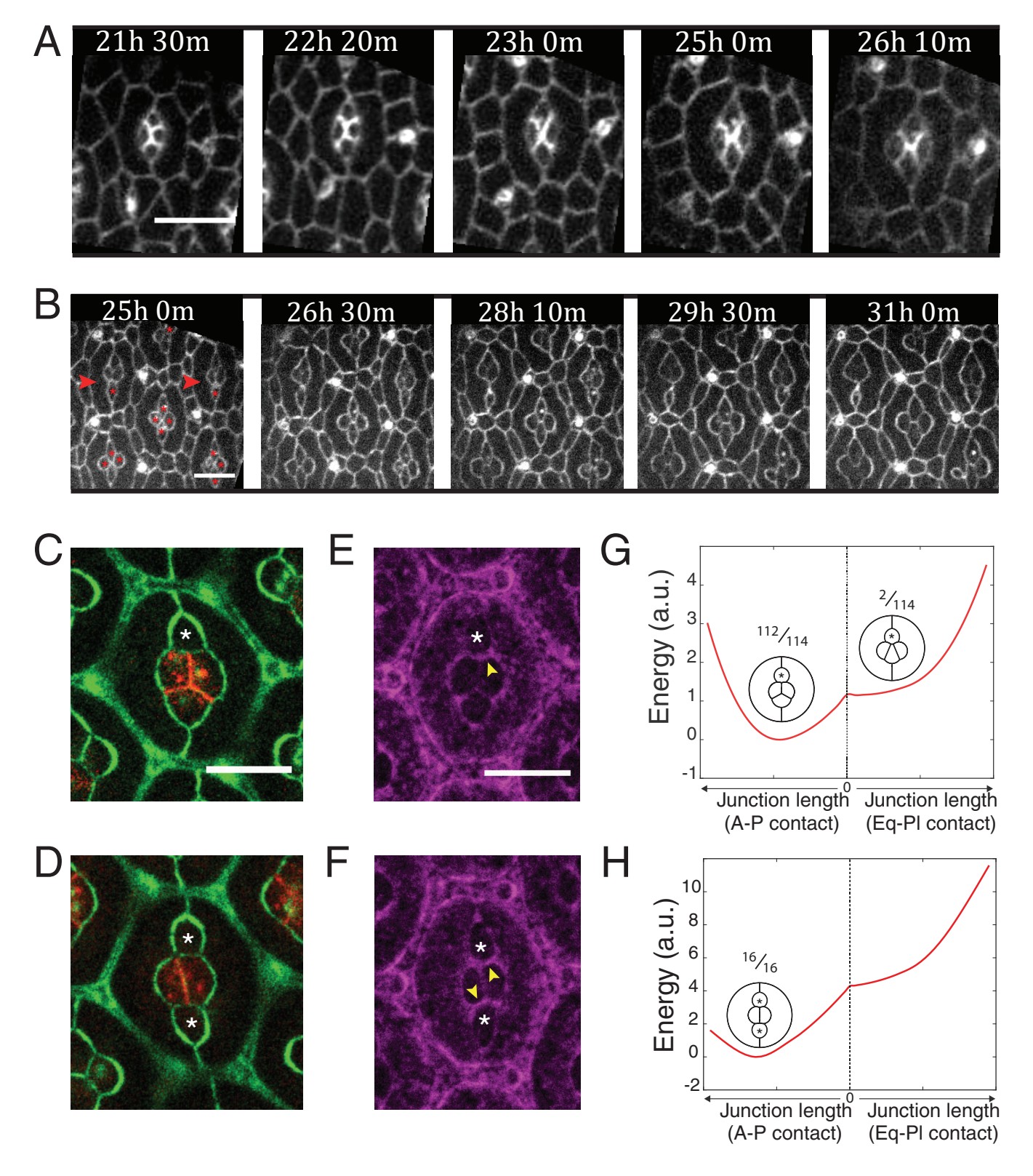

**Figure 6.** Ncad mediated MyoII contractility impacts on cone cell arrangements. (**A**) Snapshots of a movie at different APF from wildtype retina labelled with β-cat::GFP. Scale bar, 5 μm. (**B**) Snapshots of a movie at different APF from *Ncad*^*M19* mosaic mutant with Ecad::GFP and *Ncad*^*M19* cells (red asterisks). Mosaic ommatidia that failed to undergo normal cell rearrangement are indicated by red arrowheads. Scale bar, 5 μm. (**C**), (**E**) Equatorial

*Figure 6 continued on next page*

*Figure 6 continued*

*Ncad*[M19] cone cell (white asterisk) in mosaic mutant with Ecad::GFP (green) and Ncad (red) in (C) and Zip::YFP (magenta) in (E) (both (C) and (E) total n = 112). (D), (F) Image of equatorial and polar *Ncad*[M19] cone cells (white asterisks) with Ecad::GFP (green) and Ncad (red) in (D) and Zip::YFP (magenta) in (F) (both (D) and (F) total n = 16). (G) Energy profile of ommatidia with an equatorial *Ncad*[M19] cone cell as a function of the central contact length (left direction: vertical contact length, right direction: horizontal contact length). Diagrams show corresponding simulations, with occurrence numbers observed experimentally. (H) Energy profile of ommatidia with equatorial and polar *Ncad*[M19] cone cells as a function of the central contact length (left direction: vertical contact length, right direction: horizontal contact length). Diagrams show corresponding simulations, with occurrence numbers observed experimentally.

where C(E,N)|C(E,N) contacts are transformed into C(E,N)|C(E) contacts, and indeed observed increased levels of MyoII in these contacts (*Figure 6E,F*). To further test this hypothesis, we estimated the energy of the system as a function of the central junction length in both vertical and horizontal configurations (*Figure 6G,H*). Note that this required to fix that length during the minimization process. We found that the model predicts an energy minimum in the vertical configuration in both cases (when either 1 or 2 of the polar and equatorial cells are *Ncad* mutants), consistent with our experimental observations. Thus, cell mechanical properties, indirectly controlled by Ncad expression, not only impact on cell shapes but also on cell arrangement.

## Discussion

We showed that the adhesion provided by Ecad and Ncad homophilic bonds have a moderate direct contribution to interfacial tension as compared to MyoII dependent contractile forces. Our in vivo findings are consistent with in vitro measurements using the shapes of cell doublets to infer the relative contribution of adhesion and cortical tension to interfacial tension (*Maître et al., 2012*). Here we demonstrate that in vivo, the contribution of adhesion to interfacial tension is roughly half of MyoII cortical tension. Our data indicate that the hypotheses of differential contractility (*Harris, 1976*; *Brodland, 2002*) or differential adhesion (*Steinberg, 1963*) are not mutually exclusive, and the balance of contractility and adhesion determines cell shapes, cell arrangement (*Lecuit and Lenne, 2007*; *Käfer et al., 2007*; *Hilgenfeldt et al., 2008*) and cell sorting (*Krieg et al., 2008*). The moderate contribution of adhesion bonds to interfacial tension might explain why cadherin binding affinities are not predictive of cell sorting outcomes in vivo and in vitro (*Shi et al., 2008*; *Leckband and Sivasankar, 2012*).

Our work unravels a cell-scale (autonomous) and a junction-scale (non-autonomous) control of cell shape through actomyosin contractility. Following previous models of epithelial mechanics (*Käfer et al., 2007*; *Hilgenfeldt et al., 2008*), we confirm that actomyosin contractility generates a cell-scale elastic tension at the cell periphery, which restricts cell deformation. This elastic tension is likely to be dependent on the stiffness of the actomyosin network bound to the membrane (*Salbreux et al., 2012*). Our data constrain the model and reduce the number of free parameters down to one, an effective elastic constant. Our model shows that the cell-scale elasticity is crucial to stabilizing the four-cone cell arrangement and it is possible that cell elasticity also ensures correct global patterning of the retina. Analysis of our measurements of mechanical properties and quantification of molecular distribution demonstrate that MyoII contractility also contributes locally to tension at cell contacts (cortical tension) to shape cone cell arrangement. This local contribution of MyoII to tension was not considered in previous works (*Käfer et al., 2007*; *Hilgenfeldt et al., 2008*).

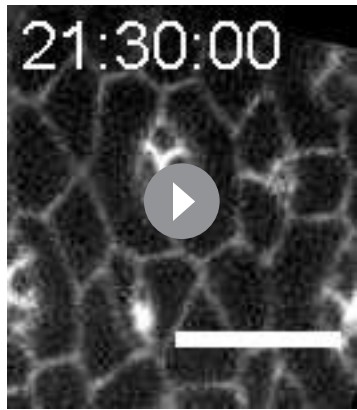

**Video 4.** A-P to Eq-Pl transition in wildtype retina. Movies starting from 21:30:00 APF. Frame rate is 10 min/frame. Labelling: β-cat::GFP. Scale bar, 5 μm.

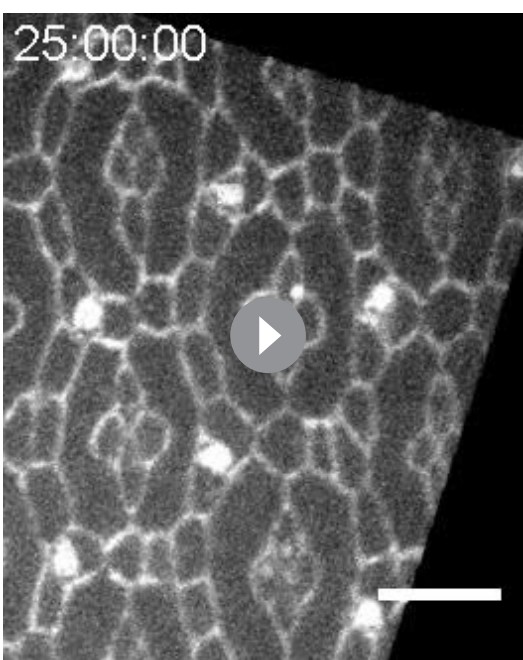

**Video 5.** Defects in cell rearrangements in *Ncad^{M19}* mosaic mutants. Movies starting from 25:00:00 APF. Frame rate is 10 min/frame. Labelling: Ecad::GFP. Scale bar, 5 μm.

The cell-scale elasticity and junction-scale cortical tension contributions are on the same order of magnitude (*Figure 5—figure supplement 2E*) and are both crucial to predicting cell shape.

MyoII distribution and thus contractility is strongly dependent on cadherins. While the role of Ecad on contractility during tissue morphogenesis is well documented (*Lecuit and Yap, 2015*), the role of Ncad is poorly known. We identified a dual role of Ncad on cell shapes and cell arrangement. Junctional N-cadherin bonds yield contact expansion between Ncad-expressing cells. However, this effect is moderate and cannot alone account for the shapes of cells in the ommatidia. Through the determination of MyoII distributions at cell contacts, we uncovered another mechanism mediated by Ncad at heterotypic cell contacts, where a low level of Ncad is detected at junctional plane (unbound). Heterotypic contacts between cells expressing Ecad and Ncad and cells expressing Ecad only exhibit increased local contractility as compared to homotypic contacts. This difference in contractility cannot be explained only by differences in adhesion contributed by both Ecad and Ncad. This is a junction-autonomous property, as in an Ecad- and Ncad-expressing cell (C), we observed increased contractility at heterotypic contacts irrespective of the other contacts of the cell (heterotypic and/or homotypic). Our data suggest that unbound Ncad has the ability to redirect MyoII at heterotypic contacts via its signaling intracellular region. Interestingly, this does not seem to be specific to the retina and might be a more general mechanism, as suggested by our observations in the larval wing disc. N-cadherin was found to polarize MyoII contractility directly through it cytoplasmic partners such as *β*-catenin (*Ouyang et al., 2013*) or indirectly through its interplay with Ecad (*Scarpa et al., 2015*), presumably through an indirect mechanism. Cadherin-mediated adhesion is tightly coupled to actomyosin through small GTPase including Rho and antagonistic Rac (*Takeichi et al., 1997*; *Ratheesh et al., 2013*). Homophilic N-cadherin dimerization activates Rho (*Comunale et al., 2007*; *Charrasse et al., 2002*; *Marrs et al., 2009*; *Taulet et al., 2009*; *Puvirajesinghe et al., 2016*) and Rac (*Matsuda et al., 2006*). Also, actin organisation has been shown to be able to affect MyoII (*Reymann et al., 2012*). We did not detect any significant variation in Rho activities among different contacts of the ommatidia using a biosensor which detects active Rho1 (*Munjal et al., 2015*)(data not shown). Further experiments will be required to resolve the mechanism by which unbound Ncad could activate MyoII.

High MyoII contractility induced by cell contact molecules at tissue boundary has a significant impact on tissue separation (*Dahmann et al., 2011*; *Major and Irvine, 2006*; *Fagotto, 2014*). In *Drosophila*, supracellular actomyosin structures are found at boundaries in wing imaginal discs (*Major and Irvine, 2006*; *Landsberg et al., 2009*; *Monier et al., 2010*; *Umetsu et al., 2014*; *Bielmeier et al., 2016*) and embryos (*Monier et al., 2010*; *Röper, 2013*; *Laplante and Nilson, 2011*). We show here that the four cone cells in ommatidia form a boundary with primary pigment cells through increased MyoII contractility at the C(E,N)|P(E) heterotypic contacts. This MyoII cable is reminiscent of that triggered by Crumbs anisotropy at the border of placodes in the *Drosophila* (*Röper, 2012*). Cells inside the placodes have higher levels of Crumbs than cells outside placodes. In the peripheral placode cells, Crumbs homophilic interactions, which are thought to negatively regulate MyoII, lead to the selective accumulation of the Myosin cable at the boundary depleted of Crumbs. One could envision that Ecad anisotropy could lead to the accumulation of MyoII at the cell contacts having a high level of Ecad. We ruled out this possibility here as we found conditions where

MyoII and Ecad anisotropy do not correlate (*Figure 2—figure supplement 1G*). In the retina, we showed that accumulation of MyoII is junction-autonomous and determined by the expression of adhesive molecules in the apposed cells.

At the heterotypic contacts, MyoII is asymmetrically distributed: it is mainly localized at the cortex of the Ecad and Ncad expressing cells. A recent study on the localization of polarity proteins on either side of cell interfaces made a similar observation (*Aigouy and Le Bivic, 2016*). From a mechanical point of view, the asymmetry of MyoII is an interesting observation as it suggests that tension can be set and modified asymmetrically. As a consequence, shrinkage or extension of a junction might be driven unilaterally from one of the two apposed cells. So far mechanical models of epithelia, including ours, do not take asymmetry into account, a property which would be interesting to explore further in the future. The adhesion molecules that are engaged in trans-bonds at cell contacts are symmetric in the apposed membranes. Thus, they cannot be the direct cause of this asymmetry. Instead, our data suggest that asymmetrically distributed unbound Ncad could signal to MyoII and cause its asymmetry. While asymmetric localization is an essential feature of planar polarity components (*Goodrich and Strutt, 2011*), it is largely unexplored for other junction constituents. It will be important to determine whether cytoskeletal components and regulators and members of adhesion complexes, also show asymmetric localization.

High MyoII contractility at contacts between two cell types might represent a general mechanism, which could be important for lineage sorting and elimination of misspecified cells (*Bielmeier et al., 2016*). Given the importance of E- to N-cadherin switch in epithelial-mesenchyme transition (*Wheelock et al., 2008*), our findings may also have implications in other developmental processes.

## Materials and methods

### Drosophila stocks and genotypes

To visualize Myosin-II in wildtype retinas, we used Zip::YFP(CPTI-100036) and $Sqh^{AX3}$ /FM7; sqh-Sqh ::GFP flies (*Karess et al., 1991*). To quantify the levels and asymmetry of Myosin-II at contacts in both $Ncad^{M19}$ mutant and misexpression background, we used Zip::YFP (RRID:DGGR_115082) and Sqh-Sqh::Cherry (*Martin et al., 2009*) as probes respectively. FRT40A, $Ncad^{M19}$ mutants and UAS-Ncad flies were gifts from Tadashi Uemura (*Iwai et al., 1997*). UAS-NcadΔcyto flies was a gift from C.H. Lee (*Yonemura et al., 2010*). UAS-Sqh$^{T20ES20E}$ flies (RRID:BDSC_64411) was a gift from R. Karess (*Jordan and Karess, 1997*). $Sqh^{Ax3}$ FRT19A/FM7 flies (RRID:BDSC_25712) are from Bloomington *Drosophila* stock centre. In laser ablation experiments, Ecad::GFP (RRID:BDSC_60584) (*Huang et al., 2009*) and β-catenin::GFP (*Huang et al., 2011*) knock-in flies used for visualizing the AJs were gifts from Y. Hong. Ncad::mKate2 flies are generated in house using the CRISPR/Cas9 technique (*Port et al., 2014*). Ncad::GFP flies are from the service of inDROSO. See belows for details of both Ncad knockin flies.

Genotypes used in experiments were as followed:

Figure 1A: *Ncad::mKate2, Ecad::GFP*
Figure 1C: *Ncad::GFP*
Figure 1D: *Ecad::GFP*
Figure 1E: *Zip::YFP/ +*
Figure 1F: *eyFLP; Ecad::GFP, FRT40A, GMR-Gal4 myr-RFP/ FRT40A, Ncad$^{M19}$*
Figure 1G: *eyFLP; Zip::YFP, FRT40A, GMR-Gal4 myr-RFP/ FRT40A, Ncad$^{M19}$*
Figure 1- figure supplement 1A: *Ncad::GFP*
Figure 1- figure supplement 1B: *Sqh$^{Ax3}$; sqh-Sqh::GFP/ sqh-Sqh::GFP*
Figure 1- figure supplement 1C: *Zip::YFP/ +*
Figure 1- figure supplement 1D: *Sqh$^{Ax3}$; sqh-Sqh::GFP/ sqh-Sqh::GFP*
Figure 2A-C: *Zip::YFP/ +*
Figure 2D-F: *eyFLP; Zip::YFP, FRT40A, GMR-Gal4 myr-RFP/ FRT40A, Ncad$^{M19}$*
Figure 2I-J: *β-catenin::GFP*
Figure 2K: *eyFLP; Ecad::GFP, FRT40A, GMR-Gal4 myr-RFP/ FRT40A, Ncad$^{M19}$*
Figure 2- figure supplement 1A-C: *Sqh$^{Ax3}$; sqh-Sqh::GFP/ sqh-Sqh::GFP*
Figure 2- figure supplement 1E: *eyFLP; Zip::YFP, FRT40A, GMR-Gal4 myr-RFP/ FRT40A Ncad$^{M19}$*

Figure 2- figure supplement 1F: *eyFLP; Ecad::GFP, FRT40A, GMR-Gal4 myr-RFP/ FRT40A, Ncad$^{M19}$*

Figure 2- figure supplement 2A-C: *eyFLP; Zip::YFP, FRT40A, GMR-Gal4 myr-RFP/ FRT40A, Ncad$^{M19}$*

Figure 2- figure supplement 2D: *eyFLP; Ecad::GFP, FRT40A, GMR-Gal4 myr-RFP/ FRT40A, Ncad$^{M19}$*

Figure 3A-D: *hsFLP; Zip::YFP/ UAS-Ncad; ActGal4, UAS-RFP/ +*

Figure 3E-G: *Sqh$^{Ax3}$; Ecad::GFP; sqh-Sqh::mCherry*

Figure 3H-J: *hsFLP; UAS-Ncad/ ActGal4 UAS-GFP, sqh-Sqh:mCherry/ +*

Figure 3- figure supplement 1A-A': *hsFLP; Ecad::GFP/ UAS-Ncad; ActGal4, UAS-RFP/ +*

Figure 3- figure supplement 1B-C: *hsFLP; UAS-Ncad/ActGal4, UAS-GFP/; Sqh::Ch/+*

Figure 3- figure supplement 2A-D: *Sqh$^{Ax3}$; Ecad::GFP; sqh-Sqh::mCherry*

Figure 3- figure supplement 2E-H: *w*

Figure 3- figure supplement 3A-C: *eyFLP; Zip::YFP, FRT40A, GMR-Gal4 myr-RFP/ FRT40A, Ncad$^{M19}$*

Figure 4A-E: h*sFLP; UAS-NcadΔcyto/ Zip::YFP; Act-Gal4 UAS-RFP/ +*

Figure 5B: *eyFLP; Ecad::GFP, FRT40A, GMR-Gal4 myr-RFP/ FRT40A, Ncad$^{M19}$*

Figure 5E-E': *Ubi-mRFP.nls, FRT19A/ FRT19A, Sqh$^{Ax3}$;; eyFLP/ +*

Figure 5F-F': *hsFLP; UAS-Sqh$^{T20ES20E}$/+; ActGal4, UAS-RFP/ +*

Figure 5- figure supplement 3A-B: Ecad::GFP

Figure 5- figure supplement 3E-F: *eyFLP; Ecad::GFP, FRT40A, GMR-Gal4 myr-RFP/ FRT40A, Ncad$^{M19}$*

Figure 5- figure supplement 4C': Ubi-mRFP.nls, FRT19A/ FRT19A, Sqh$^{Ax3}$;; eyFLP/ +

Figure 5- figure supplement 4D': hsFLP; UAS-Sqh$^{T20ES20E}$/+; ActGal4, UAS-RFP/ +

Figure 6A: *β*-catenin::GFP

Figure 6B-D: eyFLP; Ecad::GFP, FRT40A, GMR-Gal4 myr-RFP/ FRT40A, Ncad$^{M19}$

Figure 6E-F: eyFLP; Zip::YFP, FRT40A, GMR-Gal4 myr-RFP/ FRT40A, Ncad$^{M19}$

Movie 1, 3, 4: *β*-catenin::GFP

Movie 2, 5: eyFLP; Ecad::GFP, FRT40A, GMR-Gal4 myr-RFP/ FRT40A, Ncad$^{M19}$

## Genetics and immunochemistry

FLP/FRT system with eyFLP was used to create mosaic mutant tissues. Gal4-UAS system with hsFLP was used to induce targeted gene expression. 10 min heat-shock was performed 72 hr after egg deposition. Pupae were staged by collecting white prepupae and incubating at 25°C for the indicated times. Retinas were fixed in 4% of paraformaldehyde (PFA) in PBS for 20 mins, washed three times with PBS, permeabilised with PBT (PBS + 0.3% Triton x100), blocked with PBS + 10% NGS (Cat#50197Z, Life technology, CA, USA), immunostained with the indicated primary antibodies in PBS + 10% NGS at 4°C overnight and secondary antibodies for 2 hr at room temperature.

Primary antibodies used rat anti N-cadherin (DSHB Cat# DN-Ex 8 RRID:AB_528121) 1:20, rat anti E-cadherin (DSHB Cat# DCAD2 RRID:AB_528120) 1:20, mouse anti-*β*-catenin (DSHB Cat# N2 7A1 ARMADILLO RRID:AB_528089), 1:10 and mouse anti-stan #74 (DSHB Cat# Flamingo #74 RRID:AB_2619583), 1:10 (Developmental Studies Hybridoma Bank [DSHB]) and rabbit anti-Phospho-Myosin light Chain-II (Ser19) Antibody, 1:100 (RRID:AB_330248, #3671, Cell Signalling Technology, MA, USA). Secondary antibodies used were goat anti-mouse Alexa 488, goat anti-rabbit Alexa 555 and goat anti-rat/mouse Alexa 633 (1/500) (ThermoFischer Scientific, MA, USA). Fluorescence images were acquired with a Zeiss LSM780 confocal microscope with ×63, 1.4 N.A oil immersion objective. Images typically have 5–6 stacks, 0.5 μm apart.

## Time-lapse imaging of living pupal retinas

Pupae at indicated time after pupal formation were dissected and mounted on glass slides as described previously (*Corrigall et al., 2007*). Prepared samples in a temperature control chamber at 25°C were imaged using a Nikon spinning-disc Eclipse Ti inverted microscope with ×100, 1.4 N.A oil immersion objective. MetaMorph software was used and images were acquired every 10 min for 12 hr. Every image has ~10 stacks, 1 μm apart and stacks featuring the apical junctions were registered

using Fiji. Wildtype retinas live imaging was performed with $\beta$-cat::GFP flies and $Ncad^{M19}$ mosaic mutant live imaging was with Ecad::GFP flies.

## Laser ablation experiment and analysis

Laser ablation experiments were performed as previously described (*Rauzi et al., 2008*). Experiments were performed in $Ncad^{M19}$ mosaic mutants labelled with Ecad::GFP, $Ncad^{M19}$ mutant cells were differentiated from wildtype cells by RFP signal. Ablations in wildtype were performed on flies labelled with $\beta$-catenin. For C(E,N)|P(E) ablation experiments, contacts shared by equatorial or polar with primary pigments cells were used.

The recorded images of ablation were analysed in ImageJ by measuring the opening distance between vertices of the ablated junction. This opening distance was plotted over time and linear fit over the first 10 points was used to the recoil speed, which is used as an estimate of interfacial tension.

## Quantification of MyoII intensity

PFA-fixed retinas with Zip::YFP or Sqh::Ch to mark MyoII were imaged with Zeiss LSM780 confocal microscope and images were quantified by Fiji. Fluorescence signal at C(E,N)|P(E) contact can be clearly marked by ROI (generally of Linewidth 4 (0.439 μm) of the segmented 'selection' tool). Once the Line width is chosen for C(E,N)|P(E) contact same is used for the P(E)|P(E) and C(E,N)|C(E,N). To localize the P(E)|P(E) and C(E,N)|C(E,N)contacts, marked RFP signal was used (*Figure 2—figure supplement 2A*, *right panels*). Background was measured from the lowest frame of the image (~2.5 μm below from the adherens junction). Remaining stacks were summed on Z project (images were taken with 4–5 Z slices of 0.5 μm). Then, with chosen ROI junctional Myosin-II intensity at various contact type i. e. C(E,N)|C(E,N), C(E,N)|P(E), P(E)|P(E), C(E)|C(E) and C(E,N)|C(E), excluding the vertices, were measured. Mean intensity was measured using 'measure' tool of Fiji and background was subtracted from each.

## Quantification of asymmetric localization of MyoII

To determine MyoII localization with respect to cell contacts, we imaged retinas with Zip::YFP or Sqh::Ch to mark MyoII and Ecad::GFP to mark Ecad as a proxy for contact position. The images were acquired with a Zeiss LSM780 confocal microscope and quantified using Fiji. Intensity plot profiles ('Plot profile tool') for MyoII and Ecad were drawn from line segments of about 5 μm (generally of Linewidth 8 (1.05μm) of the segmented 'selection' tool) intersecting cell contacts orthogonally and at their middle. Mean intensities values were plotted for MyoII and Ecad. We used Gaussian fits to determine the position of intensity peaks and the signal to noise ratio of individual intensity line traces to estimate the precision in localization (*Bobroff, 1986*). We used multicolour Tetraspek microspheres 200 nm diameter (Invitrogen/Life Technologies, CA, USA) to measure the chromatic shift between red and green channels, which was found to be 50 and 70 nm in x, y directions, respectively.

## Angle $\theta$ measurement and ratio *Lm/Ls* measurement

The 'Angle' tool in Fiji was used to measure the angle $\theta$. The brightest pixel at the contact point was used as the angle vertex. Angles are measured for different types of cell contacts between cone cells and primary pigment cell, in wildtype as well as in $Ncad^{M19}$ mosaic conditions. The lengths are measured using the straight line 'Selection' tool of Fiji.

## Statistics

All the statistical analyzis was done in Matlab. We used the non-parametric Mann-Whitney U test on pairs and systematically applied a Bonferroni correction for multiple comparisons. Note that P-values shown in graphs include the Bonferroni correction (p>0.5, N.S). Summary for all the statistical value is in *Supplementary file 1* – table 3.

## Simulations

Simulations were performed with Surface Evolver version 2.7 (*Brakke, 1992*). Mesh grooming was implemented during minimization by refinement, and various refinement lengths have been tested

to ensure that the system had really reached energy minima. The perimeter elasticity term in the energy function (*Equation 1*) was programmed by method instance, which can be defined in the datafile. Tension was specifically set for each contact depending on its type (See parameter measurements and model simulations section).

## Parameters measurements and model simulations

Simulations of ommatidia rely on the minimization of the energy function using Surface Evolver. Surface Evolver is a freely available software (*Brakke, 1992*) designed for the study of objects maintained by surface energy (in our 2D case, line energy) and other customizable forms of energy (in our case, perimeter elasticity). Surface Evolver evolves the given surface towards its minimal energy by a gradient descent method. Area of each cell is fixed in the model, even though the apical area can change experimentally. This choice is driven by simplicity arguments. Indeed, area variations could be accounted for with an area elasticity term (in the form $K_A(A - A_0)^2$, where $K_A$ is the area elastic constant, and $A$ and $A_0$ are the actual and preferred area, respectively). Yet, and unlike perimeter elasticity, area elasticity is not crucial to select a shape or configuration (*Hilgenfeldt et al., 2008*) but mostly to set cell area. Hence, we chose to fix the area so that it matches the experimentally measured one, which spared us from having additional free parameters ($K_A$ and $A_0$). In MyoII perturbation experiments, in which cell area is significantly modified, we changed the fixed area to that measured in experiments.

The simulation parameters are $\gamma_{loc}$, which depends on the cell contact type, the elastic constant $K$, which we assume constant for all cells, and the preferred perimeters $p_0$. Using our circular ablation experiments to determine preferred perimeters, our measurements of $\gamma$ for the different contact types, and the fact that $\gamma_{loc} \approx \gamma - 2K\frac{\Delta p}{p_0}$, K is the only free parameter remaining. We ran simulations with $K$ ranging from 0.1 to 6 and fitted the resulting shapes to wildtype and Ncad mosaic ommatidia. The geometrical descriptors that we used for the fit are i) the contact angle $\theta$ between cone cells and primary pigment cells, and ii) the ratio *Ls/Lm. Ls* is the length of the junction shared by the posterior/anterior cone cell and the polar/equatorial cell, and *Lm* is the length of the junction shared by equatorial and polar cells (*Figure 5A*). To actually perform the fit, we calculated the sum of residuals for the measured angles and ratios in five configurations (one wildtype +4 different *Ncad*[M19] mosaic configurations), hence 2 x 5 = 10 residuals. We used a weighted least square method to take into account that the descriptors (an angle and a length ratio) are different quantities. Note that to simulate Ncad mosaic ommatidia, we only changed the parameter $\gamma_{loc}$ according to the contact type. For example, if the anterior cone cell lacks Ncad, then its contacts shared with equatorial and polar cone cells become C(E,N)|C(E) and its contact shared with the primary pigment cell becomes C(E)|P(E). Tensions were set according to the ablation experiments performed for each contact type.

## Estimation of the contribution of adhesion and cortical tension to $\gamma_{loc}$

Local tension $\gamma_{loc}$ results from the balance between MyoII contractility $\sigma$ and cadherin-based adhesion $\omega_N$, and we were interested in weighing their respective (direct) contributions. In order to do so, we assumed that adhesion molecules and motor molecules have an additive and antagonistic role. Hence we have $\gamma_{loc} = \sigma - \omega$. $\omega = \omega_E + \omega_N$ if both Ecad and Ncad are present at the contact, and $\omega = \omega_E$ if only Ecad is present. We assumed that $\sigma$ is proportional to MyoII intensity ($\sigma = \alpha C_M$) and $\omega$ proportional to Cadherin intensity ($\omega_E = \beta C_E$ for Ecad and $\omega_N = \delta C_N$ for Ncad). Tension measurements combined to intensity measurements provide an equation for each contact type (C(E,N)|C(E,N), C(EN)|P(E), P(E)|P(E), C(E,N)|C(E) and C(E)|C(E)), so that we have 5 equations for 3 unknowns ($\alpha$, $\beta$, and $\delta$). We use a least square fit method to find the best solution to this overdetermined system, thus estimate ($\alpha$, $\beta$, $\delta$) and consequently determine the relative contributions of MyoII ($\sigma$), Ecad ($\omega_E$) and Ncad ($\omega_N$) to $\gamma_{loc}$ for the different contact types (*Figure 5D*).

## Simulations of MyoII mutants and MyoII overexpression

MyoII manipulation experiments changed the apical areas of the cone cells and length of the cell contacts (*Figure 5—figure supplement 4A,B*). Myosin-II light chain (*Sqh*[Ax3]) mutant cone cells showed larger apical surface area than their wildtype counterparts. Cone cells misexpressing the constitutively active Myosin-II light chain (UAS-Sqh[T20ES21E]) showed smaller apical surface area than their wildtype counterparts. To simulate the shape of these perturbed cells, we measured the area

(A) of these cells to fix it in the simulations and the target perimeter by $p_0 = 2\sqrt{A\pi}$. The in silico patterns obtained for this simple change in area and target perimeter are in good agreement with the in vivo cell patterns (*Figure 5—figure supplement 4C,C', D,D', E*).

### Cell contact length measurement in ommatidium with two *Ncad^M19* cone cells

PFA-fixed retinas with Ecad::GFP and RFP to differentiate wildtype from *Ncad^M19* mutant cells were used to measure the junction length of C(E,N)|C(E,N), C(E)|C(E), C(E,N)|C(E) cell contacts in ommatidia with two adjacent cone cells *Ncad^M19* mutants. Lengths were measured using 'line tool' of Fiji. Different types of lengths measured in an ommatidium is normalized to its C(E,N)|C(E,N) length.

### Quantification of Ecad intensity

PFA-fixed retinas with Ecad::GFP and RFP to differentiate wildtype from *Ncad^M19* mutant cells. Images were obtained with Zeiss LSM780 confocal microscopy and Fiji was used for quantification. Background subtraction was not used since the background was nearly zero. Stacks were summed on 'Z project'. Linewidth 4 (0.659μm) of the segmented 'selection' tool was used to measure the mean intensity of junctional Ecad according to the contact type.

### Quantification of MyoII intensity in *Ncad^M19* mosaic ommatidia with only one wildtype Ecad and Ncad expressing cone cell

PFA-fixed retinas with Zip::YFP to mark MyoII and RFP to differentiate wildtype from *Ncad^M19* cells were imaged with Zeiss LSM780 confocal microscope and images were quantified by Fiji. Stacks were summed on 'Z project' for all the images. Background was measured from the center (apical region) of any cone cell. Linewidth 4 of the segmented 'selection' tool was used to measure mean intensity around wildtype cell and around *Ncad^M1* mutant cell. Background was subtracted from wildtype and mutant mean intensities for each image. After background subtraction, intensities were compared (wildtype n = 41, mutant n = 41).

### Quantification of F-Actin intensity

PFA-fixed retinas with Zip::YFP to mark MyoII, RFP to differentiate wildtype from *Ncad^M19* mutant cells and phalloidin staining for F-actin. Images were obtained with Zeiss LSM780 confocal microscopy and Fiji was used for quantification. Stacks were summed on 'Z project'. Linewidth 7 (0.615μm) of the segmented 'selection' tool was used to measure the mean intensity of junctional F-Actin according to the contact type (junctional Zip::YFP was used for the reference).

### Quantification of Ncad intensity

PFA-fixed retinas with Ncad::GFP were obtained with Zeiss LSM780 confocal microscopy and Fiji was used for quantification. Line width 5 (0.659μm) of the segmented 'selection' tool was used to measure the mean intensity. For each measurement at the C(E,N)|C(E,N) and C(E,N)|P(E) contacts, background is measured adjacent to the contact and subtracted from the signal at junctions.

### Analysis of localization error in Ecad or MyoII peaks

The localization precision *ΔX* of Ecad or MyoII peaks was evaluated using (*Bobroff, 1986*) $\Delta X \sim \frac{1.8}{SNR}\sqrt{\Gamma\delta x}$, where $\Gamma$ is the standard deviation of the Gaussian fit of the intensity profiles, *SNR* is the signal to noise ratio, and δx is the pixel size. Typical values were $\Gamma_{Ecad} \sim 250$ nm, $\Gamma_{MyoII} \sim 300$ nm, $SNR_{Ecad} \sim 34$ and $SNR_{MyoII} \sim 10$ and δx=131 nm. The analysis of multiple intensity profiles (n=10) led to $\Delta X_{Ecad}$ = 5–22 nm and $\Delta X_{MyoII}$ = 18–77 nm.

### Generation of CRISPR/Cas9 mediated Ncad::eGFP flies

Ncad::eGFP flies were designed and generated by *inDROSO* functional genomics (France). eGFP was inserted just before the stop codon of Ncad with a flexible linker GVG and the resulting flies was validated by sequencing. Homozygous flies are viable and occasionally exhibit islets of black cells.

## Generation of CRISPR/Cas9 mediated Ncad:mKate2 flies

### Plasmid construction

Cloning was performed with the Gibson assembly Mix (New England Biolabs, Ipswich, MA, USA). PCR products were produced with the Phusion Hot Start II HF DNA Polymerase (ThermoFischer Scientific, MA, USA). All inserts were verified by sequencing. Primers used for plasmid construction are listed in *Supplementary file 1* - table 4. Primers gRNA-NCadFw and gRNA-NCadRev were used to obtain the Ncad-gRNA from pACMAN BAC DN.CAD CH321-57H14. pCFD3 plasmid containing the U6:3 promoter (from Addgene no. 49410; *Port et al., 2014*) was used to clone annealed complementary Ncad oligo-nucleotides into the BbsI digested backbone using standards procedures to produce the following 5'-to-3' configuration: U6 promoter-gRNA-Ncad-gRNA core sequence. The construct was inserted in the attP2 site on chromosome three to generate transgenic flies (BestGene Inc., Chino Hills, CA, USA).

### Ncad::mKate2 donor plasmid production

The donor plasmid was designed to introduce a mKate2-coding sequence before the stop codon of Ncad. The exogenous sequence is flanked by homology arms of 2.31 kb (5' homology) and 1.46 kb (3' homology). The 5' homology arm contains a synonymous mutation that removes the protospacer-adjacent motif (PAM) sequence for g-RNA-NCAD to prevent mutagenesis after the integration of donor-derived sequences. The 5' and 3' homology arms were PCR amplified from genomic DNA from the clone pACMAN BAC DN.CAD CH321-57H14 using primers Ncad5'. For, Ncad5'.Rev, Ncad3'-For, Ncad3'-Rev. The mKate2 coding sequence was amplified from a mKate2-containing plasmid (*Shcherbo et al., 2009*) using the primers mKate2For and mKate2Rev. The sequences of all the primers can be found in *Supplementary file 1*- table 4. All fragments were assembled by Gibson assembly Mix into pBluescript SK(+) (Stratagene, La Jolla, CA, USA) that was digested with XhoI and NotI.

### Embryo injections

Embryos from crosses between transgenic nos-cas9 (BL 54591) virgin females and U6:3-gRNA-NCAD-expressing males were injected using standard procedures. Plasmid DNA for homologous recombination-mediated integration of mKate2 into the NCAD locus was injected at a concentration of 300 ng/μl into the nos-cas9/+;U6:3-gRNA-NCAD/+ embryos. After injection of plasmids, embryos were transferred on their coversplips to a plastic box containing wet paper towel at 25°C until they hatched as larvae. Larvae were collected with forceps and transferred to a food vial with fresh yeast, followed by culture at 25°C.

### *Drosophila* genetics and screen

Approximately 2% of the injected Nos-cas9/+; gRNA-NCAD/+ larvae survived the injection and were crossed to a *w; Sp/CyO* balancer strain. In the next generation (F1), the males were conserved at 18°C and five females were pooled for genomic extraction and PCR screen. The quality of the DNA extraction was tested with the TIO-F and TIO-R primers. The presence of mKate2 insertion in the genome was detected by PCR using the m-Kate2-Fw and m-Kate2-Rv primers. When an amplification was obtained for mKate2, 30 F1 males were crossed individually with *w; Sp/CyO* females. When the F2 generation is well developed, the F1 male was sacrificed to extract the genomic DNA and screen for the presence of mKate2. Then, the progeny of positive male was amplified and stored. To confirm that the sequences remain in-frame after the CRISPR integration, the DNA sequence surrounding the fusion was amplified by PCR using primers NCAD-F2 and mKate2R2 (*Supplementary file 1* - table 4) and checked by sequencing. The resulting Ncad::mKate2 flies are homozygous viable.

## Acknowledgements

We are grateful to Y Hong, R Karess, CH Lee, T Lecuit, AC Martin, D Pinheiro, N Tapon, T Uemura and Bloomington *Drosophila* Stock Centre for generously providing fly stocks. We thank K Brakke, for surface evolver guidance and suggestions. We thank C Chardès for development and assistance with the ablation setup. R Flores-Flores on chromatic aberration testing and JM Philippe for

excellent assistance on molecular biology. We thank members of the Lenne and Lecuit groups and B Aigouy for stimulating and useful discussion during the course of this project. We thank B Aigouy, F Graner, M Labouesse, R Levayer, P Mangeol, Q Mao, P Recouvreux, C Toret for critical comments on the manuscript. We thank L Spinelli for advices on statistical analysis. This work was supported by an FRM Equipe Grant FRM DEQ20130326509 and Agence Nationale de la Recherche ANR-Blanc Grant, Morfor ANR-11-BSV5-0008 (to P-FL). P S was supported by PhD grant from the Labex INFORM (ANR-11-LABX-0054) and of the A*MIDEX project (ANR-11-IDEX-0001–02), funded by the 'Investissements d'Avenir French Government program'. We acknowledge France-BioImaging infrastructure supported by the French National Research Agency (ANR–10–INSB-04-01, «Investments for the future»).

## Additional information

### Funding

| Funder | Grant reference number | Author |
| --- | --- | --- |
| Fondation pour la Recherche Médicale | FRM DEQ20130326509 | Eunice HoYee Chan<br>Raphaël Clément<br>Edith Laugier<br>Pierre-François Lenne |
| Agence Nationale de la Recherche | ANR-11-BSV5-0008 | Eunice HoYee Chan<br>Pierre-François Lenne |
| Agence Nationale de la Recherche | ANR-11-IDEX-0001–02 | Pruthvi Chavadimane Shivakumar<br>Pierre-François Lenne |

The funders had no role in study design, data collection and interpretation, or the decision to submit the work for publication.

### Author contributions

EHoYC, Conceptualization, Data curation, Designed and performed all the genetic experiments, IF staining, confocal microscopy, and live-imaging experiments, Formal analysis, Supervision, Validation, Investigation, Visualization, Methodology, Writing—original draft, Project administration, Writing—review and editing, Commented on manuscript; PCS, Conceptualization, Data curation, Performed laser-ablation experiments and quantified all the images and data, Formal analysis, Designed the physical model, Performed the simulations and calculations, Validation, Investigation, Visualization, Methodology, Writing—original draft, Project administration, Writing—review and editing, Commented on manuscript; RC, Conceptualization, Formal analysis, Designed the physical model, Supervision, Validation, Visualization, Methodology, Writing—original draft, Writing—review and editing, Commented on manuscript; EL, Resources, Methodology, Designed and generated the CRISPR/Cas9 Ncad::mKate2 knockin line; P-FL, Conceptualization, Resources, Formal analysis, Designed the physical model, Supervision, Funding acquisition, Validation, Investigation, Visualization, Methodology, Writing—original draft, Project administration, Writing—review and editing, Commented on manuscript

### Author ORCIDs

Eunice HoYee Chan, http://orcid.org/0000-0003-3162-3609
Pierre-François Lenne, http://orcid.org/0000-0003-1066-7506

## Additional files

### Supplementary files

• Supplementary file 1. Table 1: MyoII levels in different experiments at different contact types. Table 2: Junction length, MyoII level, Ecad level and Ncad level in different experiments at different contact types. Table 3: Statistical value for all quantifications. Table 4: Oligos used in generating CRISPR/Cas9 mediated knock-in Ncad::mKate2 flies.

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
