## [Decision Letter]

Thank you for submitting your article "Patterned cortical tension mediated by N-cadherin controls cell geometric order in the *Drosophila* eye" for consideration by *eLife*. Your article has been reviewed by three peer reviewers, one of whom is a member of our Board of Reviewing Editors, and the evaluation has been overseen by K VijayRaghavan as the Senior Editor. The reviewers have opted to remain anonymous.

The reviewers have discussed the reviews with one another and the Reviewing Editor has drafted this decision to help you prepare a revised submission.

Summary:

This work is a timely addition to older studies that focused on the mechanics of cell shape and arrangement in the fly eye. Earlier work revealed the importance of mechanical cell bond tension for tissue morphology and implicated differences of adhesion strength between different cell types as a significant factor for tissue morphology. Lenne and colleagues have now taken this a step further and looked at cell bond contractility and separated contributions from adhesion and myosin contractility to effective bond tension. They found evidence for differences of contractility at different cell bonds and a role of both adhesion and myosin mediated contractility. Interestingly, levels of N-cadherin adhesion molecules appear to down regulate myosin levels and thereby cell bond tension. This work provides an important step forward to our understanding of cell packings in the fly eye. However the present manuscript has a number of problems and shortcomings. Some of the experimental evidence presented is not sufficiently clear and the authors need to be more comprehensive in the way they present the data and provide more information. Also, the analysis of the data using a simple model of cell bond mechanics is rather murky and remains in parts unclear and unconvincing. Overall the presentation of the work needs to be significantly improved before this manuscript could be suitable for publication in *eLife*.

Essential revisions:

*Experimental Analysis*

Experiments are well controlled and quantified using excellent reagents. However, some of the analysis requires improvement or better explanation.

1) The authors fail to properly highlight a major feature of the system in the main text. As the authors show, loss of N-Cadherin leads to the accumulation of high levels of E-Cadherin at junctions that were previously high in N-Cadherin. Similar crosstalk between these two adhesion molecules is observed in many other systems. Because of this, and because of the very low levels of N-Cadherin present at C/P interfaces in the wildtype, I do not think the paper does enough to show that low level N-Cadherin functions in the absence of homotypic interactions to modify MyoII levels at junctions. While the authors are careful in their discussion to avoid suggesting that there is a direct link between N-Cadherin and MyoII, to make this clearer the authors should modify the Results section and should present the complete data set (all data points) for each type of cell-cell interface in the wildtype, N-cadherin and Squash mutant condition. In each case, it would be best to provide the full set of primary data for: E and N cadherin levels, ii) MyosinII levels, and iii) junction lengths. This would help the reader to determine, for example, the extent to which the amounts of MyoII and E-Cadherin between cells in an N-cadherin mutant are independent of junctional curvature, variance in the system, and how loss/gain of MyoII changes the lengths of different type of cell-cell junction. In this vein, I think the nomenclature is confusing. After all, there is very little N-cadherin at an EN/E interface.

2) The analysis of localized Myo2 activity is tricky and could be prone to error. The authors use three proxies: Zip, Sqh and anti-phospho-Myo. Zip and Sqh signals are more frequently used, but choosing where to measure fluorescence instensity in the images is problematic. The signal is fairly diffuse and except for CP interfaces, not noteceably concentrated at membranes. How they choose the ROI to measure will be error-prone. Likewise, the P-Myo signal is highly punctated and therefore, ROI choice will introduce variation. For example, the difference between E/E and EN/EN intensity is pretty small for P-Myo and much bigger for Sqh, yet both are supposed to be proxies of the same thing. Did the authors use a control membrane marker to guide choice of ROI (eg myr-RFP) and act as a normalization agent? Since much of the data is analyzed this way, some clearer sense of how much measurement error exists should be made and shown. They should also test how well co-imaged P-Myo, Zip, and Squ measurements correlate with one another across all interfaces.

3) The analysis shown in Figure 4 is very tricky, where the authors attempt to see if Myo2 localization is biased with respect to the membrane interface. Images shown in the figure suggest that is difficult to analyze. Since the diffraction limit of detected light is much bigger than distance across an adherens junction interface, which is 100 nm, this is a technical stretch. They aligned Myo intensity traces with respect to the supposed peak of E-cad localization, but the E-cad traces are not shown. They need to also show the traces of E-cad, centering peaks and registering Myo2. In other words, how much uncertainty is there in assigning peak position to E-cad? Another problem is the low sampling number n=10. 10 scans across same cell? How did they choose the place to transect the membrane? No description of this analysis is provided in the methods. Was it done by line scan? It should be an area scan. This analysis requires elaboration. They should do the same with N-cad and Myo2 at EN/E interfaces. If their interpretation is correct, there should be less bias in the Myo2 localization with respect to N-cad, since both are present in only one cell of the interface. They need to also scan Myo2 with E-cadherin at E/E and EN/EN interfaces. If they are right, then Myo2 will be centered. Overall, I think this is a weak line of evidence, not super crucial to the paper, and better left out or more substantiated.

4) In Figure 5 – the authors claim there is no difference in Myo2 levels at an EN/E PP interface with extracellular N-cad misexpression in one P cell. But the N of the analysis is smaller than the N of the other experiments, N=10, and there is a trending difference apparent, a slight reduction in Myo2 levels. They should increase the sample size and if the trend continues, it will become statistically significant. So the interpretation of this part is a bit iffy.

5) The statistical analysis employed throughout uses only a parametric t-test. The implicit assumptions of this test (normal distribution and homogeneity of variance) are not always met by the data as it appears in their box and whisker plots. There is often varied variance and skewed distributions that appear more like gammas. The authors should instead perform pair-wise or multi-group non-parametric tests to test their hypotheses. And since they are performing multiple pair-wise tests per experiment, a Bonferroni correction should be systematically applied.

6) Interpretation of N-cadherin effects on Myo2

The work nicely shows that N-cad affects Myo2 localization. But what about other way around – is it mutually dependent or epistatic? Since they made Sqh mutant C cells, instead of looking at catenin, did the authors look at effect on N-cadherin levels?

In Figure 4 – EN/EN CP ectopic interface does not have 2-fold lower Myo2 than the EN/E CC interface in Figure 2. Unlike the EN/EN CC interface which does. It suggests something else is also at work in affecting Myo2 localization. In Figure 4, EN/E PP interface has 33% more Myo2 than E/E PP interface. Similar to C/C situation. This is a better controlled expt to analyze.

Maybe the small amount of N-cad at C/P edges is N-cad paired to E-cad. Heterotypic pairing is known in mammalian systems. The authors claim N-cad is unpaired at these interfaces but that only assumes homotypic bonding. They should discuss and qualify.

7) The majority of MyoII in the system appears to be concentrated at three way vertices where cells meet, not along junctions as suggested. The authors should take this into consideration. This also creates problems for data analysis, since it leads to an over-estimate of the levels of MyoII at short interfaces. The authors should be explicit in explaining how they have dealt with this.

8) The authors don't comment on the effects of MyoII accumulation on one-side of a junction. On the same topic, why isnt MyoII gone from the homophilic junction EN/ENdelta in Figure 5, given that MyoII is supposed to be on the N-Cadherin side (4F), i.e. it should be subject to N-Cadherin inhibition.

*Analysis of the experiments based on the mechanical model*

A simple model for cell mechanics is proposed to capture the basic features of cell packings. The model is postulated but not really tested. There are several problems with the analysis of the data based on the model and as a consequence, the conclusions taken by the authors are not fully convincing.

1) Overall the interpretation of relative impact of contractility and adhesion is unclear and muddled. At one point the authors say the effect of adhesion on shape is mostly indirect. In the Discussion, they say it is also direct. One possibility is to re-organize the manuscript to deal with the issue near the end of the results and all at once using several lines of presented evidence. Because as it stands, they hop onto the issue sporadically as new data appears. The message is very mixed. A coherent multi-pronged approach will help in clarity.

2) A related issue about clarity: the model is based on the idea that contributions to tension gamma_loc from cadherin and from myosin are linear and additive. However, a theme of the work is the idea that cadherin influences myosin levels and myosin mediated contractile tension. This is presented in a fully coherent way. Either the system is intrinsically coupled and nonlinear. Then one cannot linearly superimpose the contributions of cadherins and myosins. Or alternatively, the system is simple and linear. Then a regulation of myosin by cadherins foes not fit in the picture. This is probably mainly problem of the presentation which is confusing.

3) With their surface evolver model, they could quantify the different contributions by comparing simulations where adhesion terms are differential and contraction terms are uniform, versus simulations where both are differential. Then compare solutions to experimental data, to ask how different is each to experiment. They could repeat the modeling where contraction is always differential, and adhesion is either uniform or differential.

4) They continually interpret together analysis of C/P cell interfaces with analysis of chimeric C interfaces or chimeric P interfaces. They are assuming that the only variable in C and P cells is N-cadherin. But that is not true – they are very different cell types, occupying different positions in space. The most highly controlled situations for their experiments is where either C cells or P cells are chimeric for either N-cadherin or Sqh. The only known variable is N-cad or Sqh. When weighing on the contributions of N-cadherin versus contactility, these are the cleaner subjects for analysis.

5) I like their analysis in Figure 3 where they find contractility has about 3/4 impact on local tension and N-cad adhesion about 1/4 impact. Another way of looking at this is to normalize laser ablation retraction speed with local Myo2 intensity for the C/C interface. When I estimated this for C/C interfaces Speed/Zip (x10^-5) is EN/EN 3.3; E/E 4.6; EN/E 3.8. Tension normalized for contractility is not identical for the three interface types, yet they would be the same if tension was only affected by differential contractility. This suggests that N-cad reduces tension at interfaces independent of its effects on Myo2, and is consistent with reducing tension through adhesion. But since non-normalized tension at E/E interfaces is two-fold more than at EN/EN interfaces, clearly contractility is also important.

6) Figure 3 shows that the energy profile has two minima, separated by a large barrier. It is suggested that the lower minimum is selected by the cells and describes the observed cell packing ("the most favorable configuration" according to the authors.). However, the principle from equilibrium thermodynamic that energy minima are selected does not apply to systems out of thermodynamic equilibrium such as tissues. Therefore both energy minima could be selected by the system. It is quite unclear given the proposed model why only one of these minima is selected. The fact that the model has two competing minima but that only one configuration is observed in WT leaves some doubts about the origins of the observed packing geometry and the validity of the model. It should be noted that the question of how the system selects the equatorial-polar cone cells in contact is very interesting. The proposed mechanism however based simply on energy minimization seems incomplete at best and could be wrong.

7) The model is based on the idea that cell area is fixed. However it is quite unclear what mechanisms keep area fixed. Furthermore, some myosin perturbations do lead to area changes as the authors report. If a fixed area is considered, it would be important to show that area is indeed fixed under at least some perturbations.

8) When looking at the supplement, "simulation" is explained to be "energy minimization" which is not a simulation. It is not explained how the energy minimization is done, how accurate it is etc.

9) The authors are quite careless with physical units. The interfacial tension γ and the elasticity K have units and the text is written as if the values of these parameters are determined. However different dimensionless ratios are discussed. This is not clearly distinguished from real values and it is sometimes unclear what dimensionless ratios are actually considered (e.g. Figure 3—figure supplement 1).

10) The key parameter gamma_loc is determined from Video 3. I would like to see a proper analysis determining this (dimensionless) parameter.

11) The authors use the term "adhesion force" to describe bond tension related to cell adhesion molecules. I have some doubts that this concept is well defined and I do not see it defined in the present paper.

[Editors' note: further revisions were requested prior to acceptance, as described below.]

Thank you for resubmitting your work entitled "Patterned cortical tension mediated by N-cadherin controls cell geometric order in the *Drosophila* eye" for further consideration at *eLife*. Your revised article has been favorably evaluated by K VijayRaghavan (Senior editor), and two reviewers, one of whom is a member of our Board of Reviewing Editors.

The manuscript has been significantly improved. Both the writing and presentation of the Results and Discussion are greatly improved over the original submission. The text is now very clear and the figures are also improved. The authors have also made good efforts to address concerns and questions. It is quite convincing that NCad influences differential myosin localization along specific interfaces, and this contributes to the differential tensions of interfaces, and ultimately sculpts the cone and primary pigment cells. This represents a significant advance of the field. There are a few remaining issues that need to be addressed before acceptance. In particular panels G and H in Figure 6 might be somewhat misleading.

Specific comment:

Figure 6 results appear misleading. The authors used the Surface Evolver to find minimal energy solutions to cell patterns constrained by genotype. But Surface Evolver is also constrained by cell neighbor relations: cell-cell contacts are not automatically rearranged unless a user "pushes" the program to do so. Thus, starting with Eq and Pl cones in contact with each other will not lead to their rearrangement into Ant – Post cone contact, no matter what parameter values are provided. Likewise, starting with Ant – Post cones in contact, as the authors did in Figure 6GH will result in a minimal solution with Ant-Post cones still in contact, no matter what "genotypes" the cells have, WT or mutant. This could mislead the reader into thinking that the initial conditions of cell neighbor relations are identical in the Figure 6GH solutions as those shown in previous model solutions (Figure 5). And therefore the reader might incorrectly conclude that the models use identical starting conditions for cell neighbor relations. The authors could delete the panels G and H, and simply show the results of Figure 6. These compare energies for different minimizations using different starting conditions. It is not surprising that the authors do not show a comparable energy profile for a wildtype ommatium because the A-P junction valley for WT may still be lower than the Eq-Pl junction valley. If true, this would further add to doubts as to the utility of applying the surface evolver method to analyzing cell-cell rearrangements.

---

## [Author Response]

*Essential revisions:*

*Experimental Analysis*

*Experiments are well controlled and quantified using excellent reagents. However, some of the analysis requires improvement or better explanation.*

*1) The authors fail to properly highlight a major feature of the system in the main text. As the authors show, loss of N-Cadherin leads to the accumulation of high levels of E-Cadherin at junctions that were previously high in N-Cadherin. Similar crosstalk between these two adhesion molecules is observed in many other systems. Because of this, and because of the very low levels of N-Cadherin present at C/P interfaces in the wildtype, I do not think the paper does enough to show that low level N-Cadherin functions in the absence of homotypic interactions to modify MyoII levels at junctions. While the authors are careful in their discussion to avoid suggesting that there is a direct link between N-Cadherin and MyoII, to make this clearer the authors should modify the Results section and should present the complete data set (all data points) for each type of cell-cell interface in the wildtype, N-cadherin and Squash mutant condition. In each case, it would be best to provide the full set of primary data for: E and N cadherin levels, ii) MyosinII levels, and iii) junction lengths. This would help the reader to determine, for example, the extent to which the amounts of MyoII and E-Cadherin between cells in an N-cadherin mutant are independent of junctional curvature, variance in the system, and how loss/gain of MyoII changes the lengths of different type of cell-cell junction. In this vein, I think the nomenclature is confusing. After all, there is very little N-cadherin at an EN/E interface.*

We thank the reviewer(s) for this constructive comment. A major feature of the system, the ommatidia, is the existence of contacts between cells differentially expressing two types of Cadherins, namely E-Cadherin and N-Cadherin. This feature makes the ommatidia an ideal system to study the mechanics of heterotypic contacts. We have now emphasized it in the main text and by revising the Result section which should highlight the main features.

In response to reviewers’ concern on the clarity of data presentation and interpretation, we now provide a complete raw dataset for E/Ncad, MyoII levels and junction length in wildtype, Ncad and Sqh mutant conditions (New [Supplementary-material SD13-data]-Table 1 and 2). In *Ncad* mosaic mutant condition, a graph showing the relationship among contact length, cadherin and MyoII is added (New Figure 2—figure supplement 1 for *Ncad* mosaic retinas with 2 mutant cells adjacent to each other and, H for all *Ncad* mosaic retinas with different combinations of *Ncad^M19^* and WT cells). In addition, we have renamed the contacts which now clarify the cell types, cone cell (C) and primary pigment cell (P), as well as the type of Cadherins, Ecad (E) and Ncad (N) expressed in the cells. We introduced and explained the new nomenclature in Figure 2. Clearer presentation of data will assist the readers from understanding the role of cadherins and MyoII in junction length/cell shapes and also resolve reviewers’ concern on the role of Ncad, in inducing MyoII at heterotypic contacts.

In fact, heterotypic contacts between cells expressing Ecad only and Ecad and Ncad exhibit increased local contractility as compared to homotypic contacts. We observed this irrespective of the type of cells in contact: wild type cone cell contacting primary pigment cell (C(E,N)|P(E)), WT cone cell contacting *Ncad* mutant cone cell (C(E,N)|C(E)) and primary pigment cell overexpressing Ncad contacting WT primary pigment cell (P(E,N+)|P(E)) ([Supplementary-material SD13-data]-Table 1).

If such increased MyoII level would be a consequence of changes in homotypic interactions of Ecad one would expect Ecad level to show a correlation with MyoII level. Figure 2—figure supplement 1 show this is not the case. In contrast, we observe change in MyoII when we manipulate Ncad (Figure 3). This is a junction-autonomous property, as in an Ecad- and Ncad-expressing cell, we observed increased contractility at heterotypic contacts irrespective of the other contacts of the cell (Figure 3—figure supplement 3). Our data suggest that unbound Ncad has the ability to redirect MyoII at heterotypic contacts, presumably via its signaling intracellular region.

To further test the generality of our observations, we also generated Ncad misexpressed clones in the larval wing pouch, a system in which cells express Ecad but not Ncad. We consistently found that the MyoII level is increased at the boundary between N+ clones and the surrounding tissue (New Figure 3—figure supplement 1, cyan arrowheads).

*2) The analysis of localized Myo2 activity is tricky and could be prone to error. The authors use three proxies: Zip, Sqh and anti-phospho-Myo. Zip and Sqh signals are more frequently used, but choosing where to measure fluorescence instensity in the images is problematic. The signal is fairly diffuse and except for CP interfaces, not noteceably concentrated at membranes. How they choose the ROI to measure will be error-prone. Likewise, the P-Myo signal is highly punctated and therefore, ROI choice will introduce variation. For example, the difference between E/E and EN/EN intensity is pretty small for P-Myo and much bigger for Sqh, yet both are supposed to be proxies of the same thing. Did the authors use a control membrane marker to guide choice of ROI (eg myr-RFP) and act as a normalization agent? Since much of the data is analyzed this way, some clearer sense of how much measurement error exists should be made and shown. They should also test how well co-imaged P-Myo, Zip, and Squ measurements correlate with one another across all interfaces.*

We are fully aware of the fact that MyoII activity measurement could be prone to error and that’s why we tested and provided quantification from all the reporters of MyoII activity (Zip, Sqh and P-Myo) we have in hands. We now include also the raw data for the quantification (corresponding source data files). As suggested by reviewer, we also co-imaged Zip::YFP or Sqh::GFP with P-Myo (New Figure 1—figure supplement 1). Note that although P-Myo antibody labelling has been validated and widely used in various studies in retinas (Warner et al., 2009, Yashiro et al., 2014, Deng et al., 2015), its staining remained punctate and fuzzy. Our P-myo staining in WT retinas (New Figure 1—figure supplement 1) is consistent with P-Myo images in Yashiro et al., 2014 Figure 4, with highest P-Myo staining in C|P and lowest in C|C. To conclude, despite slight differences in the subjunctional distribution of various MyoII reporters, we show a consistent relative difference in mean intensity comparing the different types of contacts (New Figure 1—figure supplement 1, [Supplementary-material SD13-data]- Table 1) and using the same protocol for analysis (Materials and methods, subsection “Quantification of Myoll intensity”). In wildtype ommatidia, MyoII level is higher at C(E,N)|P(E) contact than at P(E)|P(E) and the latter is higher than at C(E,N)|C(E,N). The same trend is observed in *Ncad-* mosaic mutants at which MyoII level is higher at C(E,N)|C(E) than at C(E)|C(E) and the latter is higher than at C(E,N)|C(E,N) (Figure 2).

*3) The analysis shown in Figure 4 is very tricky, where the authors attempt to see if Myo2 localization is biased with respect to the membrane interface. Images shown in the figure suggest that is difficult to analyze. Since the diffraction limit of detected light is much bigger than distance across an adherens junction interface, which is 100 nm, this is a technical stretch. They aligned Myo intensity traces with respect to the supposed peak of E-cad localization, but the E-cad traces are not shown. They need to also show the traces of E-cad, centering peaks and registering Myo2. In other words, how much uncertainty is there in assigning peak position to E-cad? Another problem is the low sampling number n=10. 10 scans across same cell? How did they choose the place to transect the membrane? No description of this analysis is provided in the methods. Was it done by line scan? It should be an area scan. This analysis requires elaboration. They should do the same with N-cad and Myo2 at EN/E interfaces. If their interpretation is correct, there should be less bias in the Myo2 localization with respect to N-cad, since both are present in only one cell of the interface. They need to also scan Myo2 with E-cadherin at E/E and EN/EN interfaces. If they are right, then Myo2 will be centered. Overall, I think this is a weak line of evidence, not super crucial to the paper, and better left out or more substantiated.*

The analysis of Myo-II localization with respect to the membrane interface is based on the localization of Ecad::GFP, which can be determined with a precision better than the diffraction limit, provided a high signal-to-noise ratio (see for example, Bobroff N, Rev Sci Ins 1986, 57:1152–1157). The new Figure 3—figure supplement 2 shows a linescan (width =1.05µm) of Ecad::GFP and Sqh::Ch orthogonal to junctions between P-C (between points a-b) and C-P (between points c-d).

First, we found that Myo-II intensity peaks always shifted toward C cells (n=10 different junctions), whatever the orientation of junctions. Given the signal-to-noise ratio of intensity linescan, the interface positions were localized with a precision of 5-22nm (New Figure 3—figure supplement 2) (Bobroff N, 1986.). MyoII peaks were localized with a precision of 18-77nm (New Figure 3—figure supplement 2). We used multicolour 200 nm diameter Tetraspeck (Invitrogen/Life Technologies) to measure the chromatic shift between red and green channels, which was found to be 50 and 70 nm in x, y directions, respectively. The distance between E-cad and MyoII peaks was found larger than the sum of the localization errors (New Figure 3—figure supplement 2). This further proves that MyoII is predominantly localized in the CCs.

We found consistent results using Starry night (Stan) as a membrane marker (New Figure 3—figure supplement 2, n=15). Scanning C|C interfaces show no significant bias of MyoII localization (Figure 3—figure supplement 2, n=15); within the experimental error, MyoII intensity is centered on the interface.

Note that although we are able to prove the existence of a spatial bias of MyoII localization towards Ncad expressing cells at heterotypic contacts, a more detailed analysis would be required to determine the positions and thicknesses of the MyoII cortices (Clark et al.,. 2013).

Detail on image acquisition and data analysis are in Materials and methods (“Quantification of asymmetric localization of Myoll”).

*4) In Figure 5 – the authors claim there is no difference in Myo2 levels at an EN/E PP interface with extracellular N-cad misexpression in one P cell. But the N of the analysis is smaller than the N of the other experiments, N=10, and there is a trending difference apparent, a slight reduction in Myo2 levels. They should increase the sample size and if the trend continues, it will become statistically significant. So the interpretation of this part is a bit iffy.*

We have now increased the sample size of the extracellular Ncad misexpression experiments (from N=10 to N= 19) and obtained the same result of no statistical difference (New Figure 4)

*5) The statistical analysis employed throughout uses only a parametric t-test. The implicit assumptions of this test (normal distribution and homogeneity of variance) are not always met by the data as it appears in their box and whisker plots. There is often varied variance and skewed distributions that appear more like gammas. The authors should instead perform pair-wise or multi-group non-parametric tests to test their hypotheses. And since they are performing multiple pair-wise tests per experiment, a Bonferroni correction should be systematically applied.*

As requested by the referee, we now use pair-wise non parametric tests (Mann-Whitney test) followed by a Bonferroni correction ([Supplementary-material SD13-data]- Table 3). We also increased the number of samples for some of the measurements (Figure 2, Figure 2—figure supplement 1, Figure 4). P-values shown in graphs include the Bonferroni correction (P>0.5, N.S).

*6) Interpretation of N-cadherin effects on Myo2*

*The work nicely shows that N-cad affects Myo2 localization. But what about other way around – is it mutually dependent or epistatic? Since they made Sqh mutant C cells, instead of looking at catenin, did the authors look at effect on N-cadherin levels?*

We did not observe any obvious change in both Ecad and Ncad levels at different contacts in Sqh mutant and active Sqh^EE^+ misexpressed. Thus, Cadherins and MyoII levels don’t seem to be mutually dependent.

The new data is now listed in Figure 5—figure supplement 4 and [Supplementary-material SD13-data]-Table 2.

*In Figure 4 – EN/EN CP ectopic interface does not have 2-fold lower Myo2 than the EN/E CC interface in Figure 2. Unlike the EN/EN CC interface which does. It suggests something else is also at work in affecting Myo2 localization. In Figure 4, EN/E PP interface has 33% more Myo2 than E/E PP interface. Similar to C/C situation. This is a better controlled expt to analyze.*

We agree with the reviewers that the change in MyoII level at interfaces (C(E,N)|P(E,N+) and (P(E)|P(E,N+)) where ectopically induced Ncad in P cell is not quantitatively comparable with their wildtype counterpart interfaces (C(E,N)|C(E,N) and (C(E,N)|P(E)). This quantitative difference is mostly likely due to the fact that when we misexpress Ncad in P cell, the amount of Ncad is much larger than in a WT cell (Figure 3—figure supplement 1’). As a consequence, the amount of bound and unbound Ncad at the modified junctions is different and not really comparable with WT. In the case of C(E,N)|P(E,N+) contact, we speculate that there will be few Ncad in C cell to make adhesion with many Ncad in P cell, resulting in comparatively little reduction of MyoII level. At P(E)|P(E,N+) contacts, more unbound Ncad is available in P cell to signal MyoII and so we see more MyoII at PP (33% more MyoII). In fact, the purpose of the experiment in Figure 3 and Figure 4 is to make clear that Ncad is able to reduce MyoII when it is engaged in adhesion and to elicit MyoII accumulation when it is unbound. We have no intention to claim those modified contacts are the same contacts as their WT counterparts. However, we would like to stress the fact that the influence of Ncad on MyoII at those modified contacts are qualitatively, if not quantitatively, similar ([Supplementary-material SD13-data]- Table 1). Also, the new nomenclature indicates ectopic Ncad as N+ and cell types (C, P) would clarify the situation.

*Maybe the small amount of N-cad at C/P edges is N-cad paired to E-cad. Heterotypic pairing is known in mammalian systems. The authors claim N-cad is unpaired at these interfaces but that only assumes homotypic bonding. They should discuss and qualify.*

Indeed, Ncad and Ecad heterophilic pairing exists in mammalian systems (Straub et al., 2011, Labernadie et al., 2017). However, in *Drosophila* pupal retinas, such interaction seems to be absent as Ecad mutant C cells containing only Ncad lose contact with P cells which have Ecad (Hayashi 2004, Figure 3). We now mention it in Introduction.

*7) The majority of MyoII in the system appears to be concentrated at three way vertices where cells meet, not along junctions as suggested. The authors should take this into consideration. This also creates problems for data analysis, since it leads to an over-estimate of the levels of MyoII at short interfaces. The authors should be explicit in explaining how they have dealt with this.*

We do not include the vertices in our measurements because vertices are points where three junctions meet. Thus, vertices in fluorescence images are expected to show higher MyoII levels than junctions even in the case of a homogeneous distribution. In addition, as the forces balance at vertices, it is very likely that MyoII at vertices do not contribute to junction tension. This being said, we don’t know why MyoII is found very concentrated in some of the vertices.

*8) The authors don't comment on the effects of MyoII accumulation on one-side of a junction. On the same topic, why isnt MyoII gone from the homophilic junction EN/ENdelta in Figure 5, given that MyoII is supposed to be on the N-Cadherin side (4F), i.e. it should be subject to N-Cadherin inhibition.*

From a mechanical point of view, the asymmetry of MyoII is an interesting observation as it suggests that tension can be set and modified asymmetrically. As a consequence, shrinkage or extension of a junction might be driven unilaterally from one of the two apposed cells. So far mechanical models of epithelia, including ours, do not take asymmetry into account, a property which would be interesting to explore further in the future. See new text lines in Discussion paragraph five.

Concerning the second point, MyoII is gone from the homophilic junction EN/ENdelta in Figure 5 (current Figure 4), the respective quantification of that is shown in Figure 5.

*Analysis of the experiments based on the mechanical model*

*A simple model for cell mechanics is proposed to capture the basic features of cell packings. The model is postulated but not really tested. There are several problems with the analysis of the data based on the model and as a consequence, the conclusions taken by the authors are not fully convincing.*

*1) Overall the interpretation of relative impact of contractility and adhesion is unclear and muddled. At one point the authors say the effect of adhesion on shape is mostly indirect. In the discussion, they say it is also direct. One possibility is to re-organize the manuscript to deal with the issue near the end of the results and all at once using several lines of presented evidence. Because as it stands, they hop onto the issue sporadically as new data appears. The message is very mixed. A coherent multi-pronged approach will help in clarity.*

To deal with the clarity issue, we moved the modeling part near the end of the Results section, in which we detail as much as possible the assumptions made and conclusions drawn.

In our Discussion, we now make clear statements about the role of adhesion molecules. Ncad can signal to MyoII, which makes it an indirect contributor to tension, but Ncad is also directly involved in reducing tension via adhesion bonds. See answer to next point (2) for more details about contributions of adhesion in the model.

The new modeling part is starting in subsection “The balance of local tension and cell scale contractility determines ommatidia shape”.

*2) A related issue about clarity: the model is based on the idea that contributions to tension gamma_loc from cadherin and from myosin are linear and additive. However, a theme of the work is the idea that cadherin influences myosin levels and myosin mediated contractile tension. This is presented in a fully coherent way. Either the system is intrinsically coupled and nonlinear. Then one cannot linearly superimpose the contributions of cadherins and myosins. Or alternatively, the system is simple and linear. Then a regulation of myosin by cadherins foes not fit in the picture. This is probably mainly problem of the presentation which is confusing.*

The rationale of the numerical model is to predict shapes from tensions at the cell contacts measured by ablations, irrespective of MyoII or Cadherin levels. Yet, local tension results from the balance between MyoII contractility and cadherin-based adhesion, and we were interested in weighing their respective (direct) contributions. To do so, we measured concentrations of adhesion and MyoII molecules in different cell contacts in which the local tension was known. Whether MyoII molecules are recruited through an Ncad feedback or any other pathway is not relevant to how they contribute to tension. Hence the weighting process does not involve the feedback between MyoII and Ncad (which could be modeled separately as a regulation network). From there, our choice of linearity was driven by simplicity, and we just assumed (as others Maitre et al., 2012) that adhesion molecules and motor molecules have an additive and antagonistic role. In the end, one can use this weighting to predict local tension, given the amount of MyoII and Cadherins measured at cell contacts.

We clarified in the text that the numerical model’s purpose is to predict shapes from a balance of local and cell-scale tension, while the purpose of the weighting process is to establish respective (direct) contributions of MyoII and Cadherin molecules to local tension. These two steps are now separated in two distinct paragraphs.

Now this can be found in subsection “The balance of cortical tension and adhesion determines local tension”.

*3) With their surface evolver model, they could quantify the different contributions by comparing simulations where adhesion terms are differential and contraction terms are uniform, versus simulations where both are differential. Then compare solutions to experimental data, to ask how different is each to experiment. They could repeat the modeling where contraction is always differential, and adhesion is either uniform or differential.*

In its current form, the rationale of the Surface Evolver model is to predict shapes from tensions at the cell contacts measured by ablation, irrespective of MyoII or Cadherin levels. Hence in Surface Evolver, local tension is not explicitly separated between contractility and adhesion. What the reviewer first suggests (weight the separate contributions of MyoII and Cadherins directly from Surface Evolver) is possible indeed, but requires adding free weighting parameters to the Evolver model. We thus felt it easier to determine the weight of adhesion and contractility separately using the simple linear system (see above), than to explore a huge parameter space in Surface Evolver, which is prone to get stuck in local minima.

What the reviewer then suggests (repeat the modeling with uniform contraction or adhesion) can still be done a posteriori using the weighting described earlier, and plugging it back into Surface Evolver with separate terms for adhesion and contractility. We did perform these comparisons with uniform/differential adhesion and contractility (see figure below). They indeed confirm that either uniform adhesion or contractility fail to mimic experiments/observations. Yet we feel that going back and forth between surface evolver and weighting might be very confusing to readers, and we decided not to include these in the paper.

Author response image 1.Simulation of an ommatidium with (**A**) differential adhesion and differential contraction (wildtype), (**B**) Uniform adhesion and differential contraction, (**C**) Differential adhesion and uniform contraction (**D**) Uniform adhesion and uniform contraction.**DOI:**
http://dx.doi.org/10.7554/eLife.22796.036

*4) They continually interpret together analysis of C/P cell interfaces with analysis of chimeric C interfaces or chimeric P interfaces. They are assuming that the only variable in C and P cells is N-cadherin. But that is not true – they are very different cell types, occupying different positions in space. The most highly controlled situations for their experiments is where either C cells or P cells are chimeric for either N-cadherin or Sqh. The only known variable is N-cad or Sqh. When weighing on the contributions of N-cadherin versus contactility, these are the cleaner subjects for analysis.*

We agree with the reviewers that C cells and P cells are fundamentally different and we should be cautious when concluding the effect of Ncad on MyoII in different cells. This point is now partly addressed by renaming the contacts to prevent confusion. In the manuscript, we have compared contacts from same type of cells, i.e. contacts among C cells in Ncad mutants (Figure 2—figure supplement 1 and [Supplementary-material SD13-data]- Table 1 and 2) and Sqh mutants (Figure 5—figure supplement 4 and [Supplementary-material SD13-data]-Table 2) or contacts among P cells in Ncad or extracellular Ncad misexpressed cells ([Supplementary-material SD13-data]- Table 1 and 2). One might argue that the misexpressing Ncad or extracellular Ncad in P cells experiments should be have been done in C cells. However, as C cells express Ncad, one will need to perform overexpression of Ncad chimera in Ncad mutant background. More sophisticated genetic manipulation, such as the MARCM technique, will have to be deployed. However, we have to emphasize that we consistently see the same trend in MyoII levels changes regardless of the cell types ([Supplementary-material SD13-data]-Table 1).

*5) I like their analysis in Figure 3 where they find contractility has about 3/4 impact on local tension and N-cad adhesion about 1/4 impact. Another way of looking at this is to normalize laser ablation retraction speed with local Myo2 intensity for the C/C interface. When I estimated this for C/C interfaces Speed/Zip (x10^-5) is EN/EN 3.3; E/E 4.6; EN/E 3.8. Tension normalized for contractility is not identical for the three interface types, yet they would be the same if tension was only affected by differential contractility. This suggests that N-cad reduces tension at interfaces independent of its effects on Myo2, and is consistent with reducing tension through adhesion. But since non-normalized tension at E/E interfaces is two-fold more than at EN/EN interfaces, clearly contractility is also important.*

The alternative method suggested by the reviewer is not completely accurate since it doesn’t take into account the elastic contribution to the tension measured by ablation. Hence we prefer to stick to our analysis.

*6) Figure 3 shows that the energy profile has two minima, separated by a large barrier. It is suggested that the lower minimum is selected by the cells and describes the observed cell packing ("the most favorable configuration" according to the authors.). However, the principle from equilibrium thermodynamic that energy minima are selected does not apply to systems out of thermodynamic equilibrium such as tissues. Therefore both energy minima could be selected by the system. It is quite unclear given the proposed model why only one of these minima is selected. The fact that the model has two competing minima but that only one configuration is observed in WT leaves some doubts about the origins of the observed packing geometry and the validity of the model. It should be noted that the question of how the system selects the equatorial-polar cone cells in contact is very interesting. The proposed mechanism however based simply on energy minimization seems incomplete at best and could be wrong.*

The referee is right to say that the system is likely out of equilibrium, and therefore might not systematically select the most energetically favorable configuration. Note that the system being very slow it can be considered quasi-static and very likely to sit at least in a local energy minimum (subsection “The balance of local tension and cell scale contractility determines ommatidia shape”). How the minimum is selected in the course of retinal morphogenesis is not solved, and most likely relies on the history of the system, which we barely looked at in the paper. As an example, cone cells undergo a T1 transition during retinal morphogenesis, hence transiting from one minimum to another. How this transition occurs is in itself another project that remains to be addressed.

We therefore removed the Discussion concerning the energy minimization and the corresponding subfigures (current Figure 5). We still show energy landscapes in Figure 6, where we analyze the transition defects, as they are strongly asymmetric with either only one stable minimum, or two minima with on being barely stable (subsection “Myosin-II localization mediated by N-cadherin regulates cell arrangement”).

*7) The model is based on the idea that cell area is fixed. However it is quite unclear what mechanisms keep area fixed. Furthermore, some myosin perturbations do lead to area changes as the authors report. If a fixed area is considered, it would be important to show that area is indeed fixed under at least some perturbations.*

In experiments, area is not fixed and can vary with MyoII levels, in a range which is likely to be determined by fixed 3D volume constraint and cell elasticity. Yet in the model, the area is fixed by a Lagrange multiplier. This choice is driven by simplicity arguments. Indeed, area variations could be accounted for with an area elasticity term (in the form k_a*(A-A0)^2). Yet, and unlike perimeter elasticity, area elasticity is not crucial to select a shape or configuration (Hilgenfeldt et al., 2008) but mostly sets cell area. Hence we chose to fix the area so that it matches the experimentally measured one, which spared us from having additional free parameters (k_a and A0). In MyoII perturbation experiments, in which cell area is significantly modified, we changed the fixed area to that measured in experiments.

Revised text: subsection “The balance of local tension and cell scale contractility determines ommatidia shape”, paragraph two and Material and methods subsection “Parameters measurements and model simulations”.

*8) When looking at the supplement, "simulation" is explained to be "energy minimization" which is not a simulation. It is not explained how the energy minimization is done, how accurate it is etc.*

The topology of the surface is defined in a datafile, then Surface Evolver evolves the surface towards minima of energy using a gradient descent method. Hence, the modeling process searches for energy minima in which we assume the system settles. A consequence is that the model has no time scale, but only predicts equilibrium configurations. Nonetheless, we think it could still be called simulations, in the sense that the model *simulates* an equilibrium shape under given modeling assumptions. Note that other papers modeling these types of systems with minimization techniques also used the “simulation” terminology (Kafer et al., 2007, Hilgenfeldt et al., 2008).

Revised text: Material and methods subsection “Parameters measurements and model simulations”.

*9) The authors are quite careless with physical units. The interfacial tension γ and the elasticity K have units and the text is written as if the values of these parameters are determined. However different dimensionless ratios are discussed. This is not clearly distinguished from real values and it is sometimes unclear what dimensionless ratios are actually considered (e.g. Figure 3—figure supplement 1).*

First of all it should be noted that we do not use any absolute physical values for our parameters. This is justified by the fact that ablation experiments only provide relative values of tension. From there, our (arbitrary) choice was to normalize tensions by *γ*_C|C_, so that all tensions become dimensionless and in units of *γ*_C|C_. This also sets the scale of our elastic constant K, which also has the dimension of *γ*, and is therefore also in units of *γ*_C|C_. The ratios used in figures mentioned by the reviewer were indeed confusing. Hence, we removed all these ratios and only used K in these figures (Figure 5—figure supplement 2). We now clarify in the text that K is in units of *γ*_C|C_.

Revised text in the subsection “The balance of local tension and cell scale contractility determines ommatidia shape”.

*10) The key parameter gamma_loc is determined from Video 3. I would like to see a proper analysis determining this (dimensionless) parameter.*

We haveγloc≈γ−2KΔPP0.γwas measured for each contact type (C|C, C|P and P|P) from ablation experiments. We also estimated ΔPP0 with circle ablation experiments (as shown in Video 3). We were left with the elastic constant K as the only free parameter. To determine it, we varied K (and accordingly calculated *γ_loc_* for each contact type using that(γloc≈γ−2KΔPP0) and each time we performed a simulation. We then used a least square fit method to fit simulated shapes to experimental ones, and hence determine the best K and corresponding *γ_loc_*’s (Figure 5—figure supplement 2).

We clarified this in text (subsection “The balance of local tension and cell scale contractility determines ommatidia shape”) and Material and methods subsection “Parameters measurements and model simulations”.

*11) The authors use the term "adhesion force" to describe bond tension related to cell adhesion molecules. I have some doubts that this concept is well defined and I do not see it defined in the present paper.*

Indeed we meant the contribution of adhesion bonds to interfacial tension. We now have removed the term adhesion force, which was not defined.

[Editors' note: further revisions were requested prior to acceptance, as described below.]

*Specific comment:*

*Figure 6 results appear misleading. The authors used the Surface Evolver to find minimal energy solutions to cell patterns constrained by genotype. But Surface Evolver is also constrained by cell neighbor relations: cell-cell contacts are not automatically rearranged unless a user "pushes" the program to do so. Thus, starting with Eq and Pl cones in contact with each other will not lead to their rearrangement into Ant – Post cone contact, no matter what parameter values are provided. Likewise, starting with Ant – Post cones in contact, as the authors did in Figure 6GH will result in a minimal solution with Ant-Post cones still in contact, no matter what "genotypes" the cells have, WT or mutant. This could mislead the reader into thinking that the initial conditions of cell neighbor relations are identical in the Figure 6GH solutions as those shown in previous model solutions (Figure 5). And therefore the reader might incorrectly conclude that the models use identical starting conditions for cell neighbor relations. The authors could delete the panels G and H, and simply show the results of Figure 6. These compare energies for different minimizations using different starting conditions. It is not surprising that the authors do not show a comparable energy profile for a wildtype ommatium because the A-P junction valley for WT may still be lower than the Eq-Pl junction valley. If true, this would further add to doubts as to the utility of applying the surface evolver method to analyzing cell-cell rearrangements.*

We agree with this comment and have removed Figure 6. Indeed, the results in Figure 6 are more informative and are not misleading, as the energy profiles are determined for configurations that are depicted in the figures.